# Teleultrasound in obstetrics: A systematic review and meta-analysis

Jack Le Vance [1,2]*, Matthew Vaughan [3], Tanvi Bhatia [4], Leo Gurney [2], Victoria Hodgetts Morton [1,2], R. Katie Morris [1,2]

1 Department of Applied Health Sciences, School of Health Sciences, College of Medicine and Health, University of Birmingham, Birmingham, United Kingdom, 2 Birmingham Women's and Children's NHS Foundation Trust, Birmingham, United Kingdom, 3 University of Birmingham Medical School, Birmingham, United Kingdom, 4 South Warwickshire University NHS Foundation Trust, Warwick, United Kingdom

* j.hamer@bham.ac.uk

## Abstract

### Background

Ultrasound is a common diagnostic modality in obstetrics to evaluate the fetal condition, frequently used in pregnant women classifying as high-risk. Modifications to guidelines, implementation of national initiatives, combined with an aging obstetric population has led to an increased number of high-risk patients. This places a substantial strain on outpatient obstetric services to accommodate the increased demand for serial antenatal ultrasound scans.

Recent advancements in digital technology have enabled the swift innovation of teleultrasound development. The recent pandemic has also substantially influenced technological development, as obstetric services considered alternative solutions to healthcare provision standards. This review aims to assess whether teleultrasound is feasible, acceptable, diagnostically accurate, and cost-effective for antenatal care.

### Methods and findings

We searched MEDLINE, Embase, Cochrane Database of Clinical Trials (CENTRAL), Web of Science, and PubMed databases from inception to December 2025. Primary research studies evaluating the feasibility, diagnostic accuracy, clinical utility, educational utility, acceptability, and economic viability of antenatal teleultrasound usage were included. Random effects meta-analysis was used, and results were reported as pooled proportions or risk ratio (RR) with 95% confidence interval (CI). Diagnostic accuracy was further assessed using a hierarchical summary receiver operating characteristic model.

Of the 6,561 papers screened, 71 studies (60 clinical observational studies, five qualitative studies, four economic evaluation studies, and two randomized controlled trials) were included. Image transfer was feasible for both synchronous and

which permits unrestricted use, distribution, and reproduction in any medium, provided the original author and source are credited.

**Data availability statement:** All relevant data are within the manuscript and its Supporting information files.

**Funding:** The author(s) received no specific funding for this work.

**Competing interests:** The authors have declared that no competing interests exist.

**Abbreviations:** AI, artificial intelligence; AUC, area under the curve; CENTRAL, Cochrane Database of Clinical Trials; CHD, congenital heart disease; CI, confidence interval; GRADE, grading of recommendations assessment, development, and evaluation; HSROC, hierarchical summary receiver operating characteristic; POCUS, point-of-care ultrasound; PRISMA, Preferred Reporting Items for Systematic Reviews and Meta-Analysis; RR, risk ratio.

asynchronous teleultrasound transmission, in a wide range of settings. Adequate technological infrastructure, including appropriate bandwidth and framerate requirements were vital factors for sufficient image quality and minimizing transmission delays. Visualizing gross fetal and placental structures using teleultrasound was frequently high; however, more specialized anatomy such as cardiac and neurological demonstrated lower visualization rates. Overall meta-analysis of 20 anatomical structures demonstrated teleultrasound is non-inferior at identification versus the reference standard RR 1.02 (95% CI [1.00,1.03]; $n = 4$ studies). Pooled diagnostic accuracy demonstrated excellent performance, with an AUC of 0.93 ($n = 8$ studies). The overall sensitivity was moderate at 0.70 (95% CI [0.44,0.84]), with a low false positive rate of 0.03 (95% CI [0.01,0.12]). There was evidence of educational and clinical utility for obstetric teleultrasound, particularly with novice users, demonstrating improved access to care in rural areas and low- and middle-income countries. Patient-operated telesonography demonstrated feasibility and high acceptability for performing basic fetal assessments. Three-dimensional, four-dimensional, and robotic teleultrasound did not highlight superiority to two-dimensional scanning. Patients and provider acceptability was high, citing benefits in relation to satisfaction, confidence, economic savings, and balancing healthcare equity. Teleultrasound implementation costs can be high, but were frequently accrued due to monthly savings. High-quality studies were underrepresented, suggesting a need for further research. The reporting of clear methodological and technological capabilities of the teleultrasound systems represent the main limitations, proving difficulty to replicate studies adequately.

## Conclusion

This review demonstrated the potential applicability and value of obstetric teleultrasound. This novel care model is everchanging and new devices/systems capable of telesonography are of clinical and scientific relevance. Presently, additional high-quality evidence is required, particularly using teleultrasound in a clinical context, whilst ensuring sufficient methodological detail and consistent outcome reporting.

## Author summary

### Why was this study done?

- Advancements in digital medical technology combined with the increased access to smartphones, virtual video communication, and wireless internet have enabled the swift innovation of obstetric teleultrasound development.

- Despite the rapid developments of digital teleultrasound innovations in obstetrics, the collation of available evidence, particularly assessing feasibility, diagnostic accuracy, acceptability, and economic burden is minimal and thus requires thorough exploration.

## What did the researchers do and find?

- We conducted a systematic review and meta-analysis of 71 studies across 27 countries. Teleultrasound image transfer was feasible for a range of teleultrasound technologies. The identification of fetal structures using teleultrasound were frequently non-inferior compared with conventional ultrasound for a range of clinical indications, whilst the diagnostic accuracy of obstetric teleultrasound was high.

- Teleultrasound usage by novice users with associated tele-mentoring has clinical and educational utility, particularly in low- and middle-income countries whereby the current accessibility of ultrasound can be improved.

- Patient-operated teleultrasound is a promising novel modality. However, currently, only a basic fetal assessment can be achieved. three-dimensional, four-dimensional, and robotic teleultrasound have limited scope and superiority over conventional ultrasound.

- Initial overall costings of teleultrasound may be high but frequently reclaimed back due to savings when compared with standard antenatal care. Acceptability was high for both patients and service providers.

## What do these findings mean?

- This review has demonstrated the potential applicability and value of teleultrasound for obstetrics across a range of outcomes. This novel care model is everchanging and new ultrasound devices capable of telesonography are of clinical and scientific relevance.

- Currently included literature was of high risk of bias and limited in the reporting of clear methodological detail, consistent outcome reporting, and technological capabilities of the teleultrasound systems.

- The reporting of essential regulatory and technological standards such as data encryption, patient confidentiality, liability in cross-jurisdictional teleultrasound use, bandwidth requirement, and transmission delays/failures is required to ensure a global assessment of obstetric teleultrasound functionality.

## Introduction

Ultrasound is a commonly used diagnostic modality in obstetrics to evaluate the fetal condition, which is frequently used in pregnant women who are classified as high-risk. Following recent guideline changes combined with an aging obstetric population, a greater proportion of women are now classified as high-risk, placing a strain on current obstetric services to accommodate the increased demand for ultrasound examination [1]. Advancements in digital medical technology combined with the increased access to smartphones, virtual video communication, and wireless internet have enabled the swift innovation of teleultrasound development [2,3]. The recent pandemic has also had a substantial influence on technological development, as obstetric services now need to consider alternative solutions to current healthcare provision standards.

Teleultrasound/tele-sonography and remote portable ultrasound devices combine the usage of ultrasound with telecommunication to enable remote examination either in synchronous or asynchronous format. Synchronous communication involves real-time feedback commonly conducted between a non-expert sonographer and an imaging expert [4]. Ultrasound examination with delayed transmission for expert review demonstrates asynchronous communication [5]. Obstetric teleultrasound represents potential advantages with improved patient autonomy, education, reduced hospital attendance, and economic burden [3,6,7]. Importantly, teleultrasound presents opportunities to improve overall access to ultrasound services and aims to address current health outcome disparities, particularly seen within low- and middle-income countries [6,8].

Teleultrasound usage in other healthcare settings, such as emergency medicine has been clearly demonstrated [4]. Despite the rapid developments of digital teleultrasound innovations in obstetrics, the collation of available evidence is minimal and requires thorough exploration. This review aims to assess whether teleultrasound is feasible, acceptable, diagnostically accurate, and cost-effective for the assessment of obstetric patients.

## Methodology

### Study registration

This systematic review adhered to the Preferred Reporting Items for Systematic Reviews and Meta-Analysis (PRISMA) guidelines (S1 Table) [9,10]. A protocol for this review was registered with PROSPERO (CRD42024615570).

### Inclusion and exclusion criteria

Complete inclusion and exclusion criteria can be seen within Table 1. Principally, all primary research studies regarding antenatal teleultrasound usage within obstetrics were included. Included studies assessed teleultrasound services, remote patient- or clinician-operated ultrasound devices. However, studies using remote ultrasound devices were only included if the study detailed an additional telemedicine review for the completed ultrasound images/recordings. Synchronous and/or asynchronous telemedicine transmission occurring between hospitals, between pre-hospital sectors and the hospital, or between the home and hospital were included. Clinical outcomes related to feasibility, diagnostic accuracy, clinical and educational utility, qualitative and economic outcomes were eligible for inclusion. Conference abstracts, communication papers, editorials, systematic reviews, narrative reviews, and single case reports. were excluded. Studies referring to overall telemedicine models, however, not specifying teleultrasound usage were excluded.

**Table 1. Inclusion and exclusion criteria.**

| | Inclusion | Exclusion |
|---|---|---|
| Publication type | • Full-text articles<br>○ Any of: randomized controlled trial, feasibility study, prospective/retrospective observational study, validation study, pilot study, diagnostic accuracy study, case series (two or more patients), qualitative study, economic evaluation study. | • Any of: conference abstract, communication paper, letters to the editor/editorials, systematic review, narrative review, single case reports. |
| Participants | • High or low risk pregnant women in the antenatal period. | • Any of: pregnant women in the intrapartum or postnatal period, simulated patients, non-pregnant women, neonates, infants. |
| Setting | • Any of: hospital ward or departments, community-based (general practice, rural clinic, community sonography clinics), patients' home. | • Hospital labor ward. |
| Intervention | • Any form and indication for an obstetric antenatal ultrasound scan whereby teleultrasound services or remote ultrasound monitoring were performed. However, remote ultrasound monitoring must have the images and/or videos sent for review via telemedicine. Additionally, studies either using asynchronous or synchronous transmission whereby the sonographer and the reviewer are both involved in the provision of obstetric services. | • Sonographer and reviewer are not involved in the provision of obstetric services. Non-obstetric scans or obstetric scans not performed within the antenatal period. No usage of teleultrasound transmission. Ultrasound scans performed and additionally interpreted by the same healthcare professional. Studies delineating the integration of telemedicine models, however, not specifying the usage of ultrasound services. |
| Comparator | • Not required to have a comparator. If a study has a comparator than to be compared against a reference standard such as routinely implemented antenatal care. | |
| Outcomes | • Outcomes relating to the provision of antenatal obstetric teleultrasound. Any of: clinical outcomes, economic outcomes, qualitative outcome. | • Outcomes that are not relevant to obstetric teleultrasound. |

## Search strategy, study selection, and data extraction

MEDLINE, Embase, Cochrane Database of Clinical Trials (CENTRAL), Web of Science, and PubMed databases were searched from inception to 31st December 2025. Search terms and functions were appropriately adapted for each database with medical subject headings and text words were used. The research strategy for MEDLINE and Embase can be seen in S2 Table. Reference lists of primary studies and reviews were additionally screened to identify gray literature citations eligible for inclusion. No language restrictions were applied. Covidence software (Veritas Health Innovation M, Australia) was used to manage title, abstract, and full manuscript screening. This process was repeated for the subsequent data extraction. All titles, abstracts, and full texts were reviewed independently by two out of three available researchers (JLV, MV, TB). Any conflicts were discussed with a fourth independent author (LG). For eligible studies, data was extracted into a pre-designed electronic spreadsheet. Data extracted from each study included the author, year of publication, study location, patient demographics, sample size, indication of teleultrasound usage, gestational age of usage, type of technology used, teleultrasound technical details, teleultrasound method (synchronous or asynchronous), teleultrasound performer, teleultrasound examiner, and additional study-specific outcomes. Data extraction was performed by two out of three available independent researchers (JLV, MV, TB). Any discrepancies were resolved by consensus.

## Risk of bias and trustworthiness

For observational studies, the methodological quality of these studies was assessed using an adapted form of the QUADAS-2 checklist from Whiting and colleagues and Marsh-Feiley and colleagues [4,11] (S3 Table). Seven domains, adapted for obstetric studies, were assessed to determine bias: 1) Generic quality standards; 2) Participant selection bias; 3) Index test; 4) Reference standard; 5) Flow and timing; 6) Telemedicine/technology-specific items; 7) Overall applicability. Risk of bias for qualitative and economic evaluation studies was performed using the critical appraisal checklists from the Joanna Briggs Institute (JBI) [12,13]. The Cochrane risk of bias tool (RoB2) was used to assess bias in randomized interventional studies [14]. Risk of bias were independently assessed by two out of three available reviewers (JH, MV, TB). Any discrepancies with bias assessment were arbitrated with a fourth independent author (LG).

## Certainty of evidence

The grading of recommendations assessment, development, and evaluation (GRADE) approach was used to assess the certainty of the body of evidence for each applicable outcome evaluated in the findings and presented in tabulated format [15,16].

## Data analysis

Single variable proportional data such as the identification rates of fetal/placental structures was calculated using R version 4.5.1 [17]. Random effects models were used to meta-analyze estimates of the overall proportional data and 95% confidence intervals (CIs), due to the high likelihood of statistical heterogeneity. The inverse variance method with logit transformation of the rate was used. Summary proportions and CI were configured with forest plots, including quantitative assessment for heterogeneity using $I^2$ [18].

To evaluate the diagnostic accuracy of teleultrasound compared with the reference standard, a hierarchical summary receiver operating characteristic (HSROC) model was employed. This approach allows for the simultaneous estimation of study-level sensitivity and specificity while accounting for between-study heterogeneity and when a range of thresholds are used [19]. All analyses were conducted in using R version 4.5.1 using the mada packages [17,20]. Study-level 2 × 2 contingency data (true positives, false positives, false negatives, and true negatives) were extracted and entered into R to compute individual study estimates of sensitivity and specificity with corresponding 95% CIs. The Reitsma bivariate random-effects model, a function from the mada package was used to fit a bivariate random-effects model to the

PLOS Medicine

diagnostic accuracy data to derive pooled estimates of sensitivity and specificity, along with the summary receiver operating characteristic (SROC) curve and the area under the curve (AUC) [21].

Review Manager 5.4 (RevMan) was pragmatically used to meta-analyze pairwise data, such as identification rates of fetal/placental structures in teleultrasound versus the denoted reference standard [22]. Inclusion of two of more studies with identical clinical outcomes were eligible for meta-analysis. Dichotomous outcome data were assessed with 2 × 2 tables constructed to calculate risk ratios (RRs) and 95% CI. Mantel–Haenszel method random-effects models were utilized for meta-analysis. Summary RR were presented using forest plots, allowing visual inspection and quantitative assessment for heterogeneity using $I^2$. Zero-cell adjustments to 0.5 were used as required. Meta-analysis of clinical and technological outcome data was not possible due to inconsistent outcome reporting and wide methodological variance in included studies. Therefore, a descriptive synthesis of data alongside available meta-analyzable data is therefore provided. Standardized metrics provided by included studies, such as $p$ values, have also been integrated into the results and summary tables below. Due to the small number of included studies for each meta-analyzed outcome, subgroup analysis, and publication bias was not examined.

## Results

The database and gray literature searches identified 6,561 abstracts. A total of 2,173 duplicate abstracts were removed, leaving 4,388 abstracts for screening. A total of 149 full texts were reviewed, from which 71 papers (60 clinical observational studies, five qualitative studies, four economic evaluation studies, and two randomized controlled trials) recruiting across 27 countries were eligible for inclusion (Fig 1) [23–93]. The most common reason for exclusion during full text screening was due the wrong publication type, whereby the acquired full text frequently referred to a published conference abstract without a full manuscript available. Excluded studies following full-text screening are presented in S4 Table.

### Study characteristics

All articles were published between 1995 and 2025, with the majority occurring between 2020 and 2025. Recruitment from low/middle income countries represented 14% of the total included studies [40,41,50,52,79,81,84–86,90]. One study recruited in each of: Argentina, Chile, Colombia, Ethiopia, Ghana, Indonesia, Italy, the Netherlands, Nepal, Norway, Romania, and Taiwan. Two studies recruited in each of: Egypt, French Guiana, Germany, Ireland, Kenya, and Peru. Three studies recruited in each of: Australia, Brazil, Canada, and France. Four studies recruited in Spain, six in Japan and Israel, 13 in the United Kingdom (UK), and 22 in the United States of America (USA). Four studies recruited participants in more than one country, ranging from two to 12 [26,83,86,90]. A graphical representation of study recruitment can be seen in Fig 2.

### Risk of bias

Risk of bias assessments for all included articles are presented in S5 Table. Summary results for each of the 60 observational studies, including pooled results stratified by the individual risk of bias domain using the modified QUADAS-2 checklist are presented in Figs 3 and 4. The overall quality of the articles was generally poor, with high risk of bias. This was primarily attributed to the suboptimal quality of methodological reporting regarding participant inclusion/exclusion criteria and detailed technical descriptors/standards of the teleultrasound modality used. Many studies lacked a reference group or comparison, leading to insufficient detail to enable study replication. Most commonly, observational studies failed to provide information regarding security measures and cost implications. When provided, details were frequently limited to brief statements such as encrypted/personal communication networks [33,41,45,47–49,55,84], de-identified image transfer [29,62,90], and secure teleultrasound technology [44,54]. Observational studies presenting cost outcomes ranged from detailed economic evaluations [29,54] to crude cost estimations [35,36,85], but most commonly demonstrated only short, brief statements regarding teleultrasound costings [40,45,60,75,77,86].

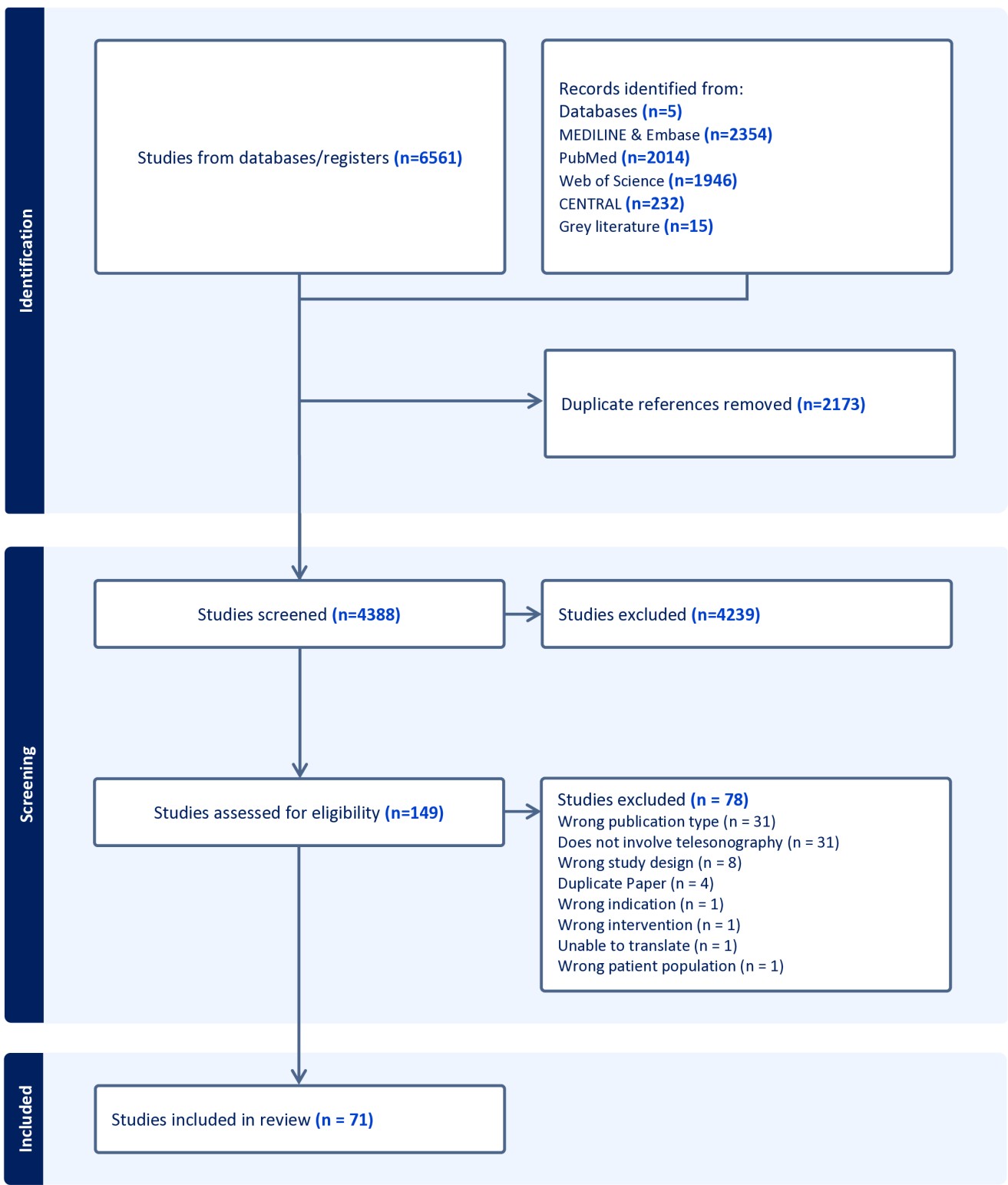

**Fig 1. PRISMA flow diagram.** CENTRAL, Cochrane Database of Clinical Trials.

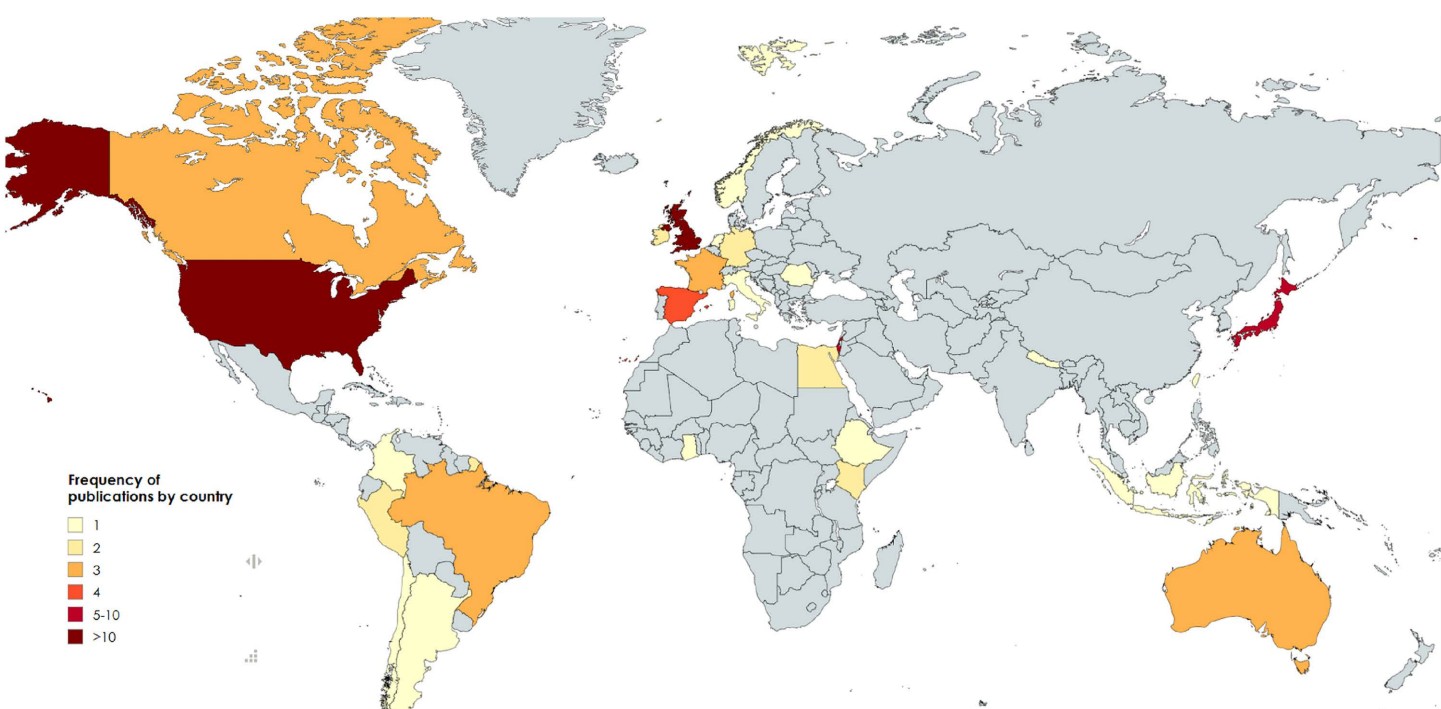

**Fig 2. World map graphical representation of included studies recruitment origin.** Produced with permission from https://www.mapchart.net/world.html.

Economic evaluation and qualitative studies were generally of lower bias risk across all categories assessed within the JBI frameworks. The two randomized interventional studies generally demonstrated low bias risk across all domains of the Cochrane RoB2 tool; however, aspects of bias concerns were raised within one study over deviations from the intended intervention and the outcomes measured (Figs 5 and 6) [65].

## Study design and methodology

Included studies were highly heterogenous in design, methodology and the outcomes measured. Study sample sizes ranged from 2 to 106,674 participants; however, 67% of studies included 100 participants or fewer. Ten% of studies incorporated tele-transmission of transvaginal ultrasound scans, whilst the remaining solely used transabdominal ultrasound scans [68,75,78,79,83,90]. Isolating results specifically for tele-transvaginal scans was not possible. Gestational age at which teleultrasound occurred was widely dispersed in the studies. Four studies assessed in the first trimester [40,68,78,79], eight in the second trimester [25,37,41,60,61,86,88,90], six in the third trimester [27,48,67,85,87,89], three in the first and second trimesters [31,65,66], 19 in the second and third trimesters [7,23,32,33,36,49,50,55,62,63,70,71,73,74,76,80,84,91], 12 in all trimesters [24,29,30,42,44,51,53,58,82,83,92,93], and 19 did not disclose [7,26,28,34,35,38,39,43,45,52,54,59,64,69,72,75,77,81]. Sixty studies assessed hospital to hospital teleultrasound transmission, four evaluated pre-hospital/community to hospital transmission [24,50,52,81], and seven studies assessed home to hospital transmission [44,65,85,87,89,92,93].

Observational studies were broadly categorized by the authors (JLV and LG) into two main subtypes. Firstly, feasibility studies considered the capabilities and specifications of different teleultrasound technologies to transmit and receive information that was deemed clinically useful [24,26–28,34,35,39–41,43,45,48,49,51,54,57,63,66–70,72–75,77,80,81,86,87,89,92]. Secondly, diagnostic accuracy studies evaluated the diagnostic capabilities of teleultrasound [23,25,29,32,33,36,37,42,44,47,51,53,55,56,59,60,62,71,78,79,83–85,88,90,91,93].

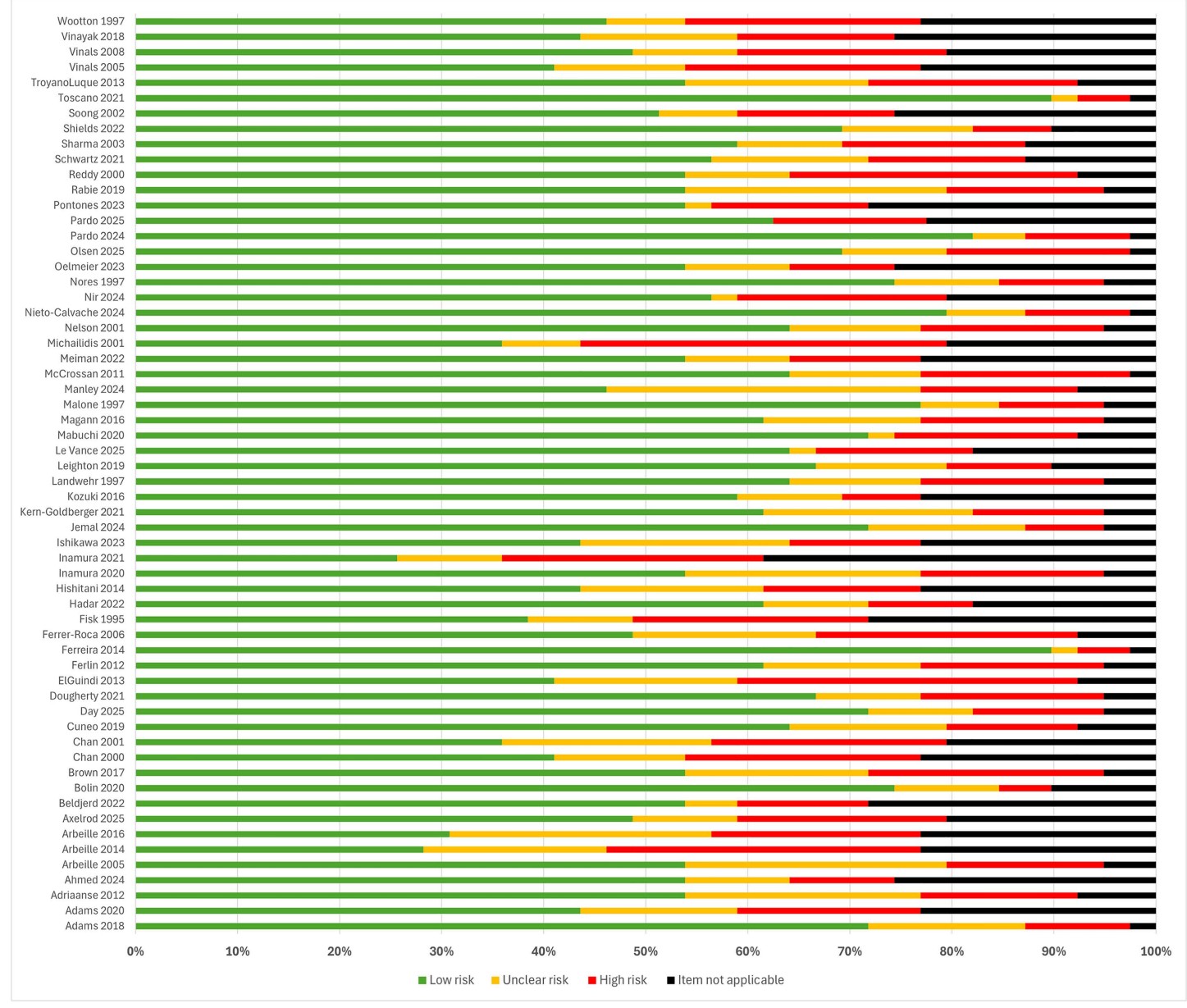

**Fig 3. Summary risk of bias results for clinical observational studies using the QUADAS-2 checklist.**

All studies either incorporated asynchronous [23,25,26,29,30,33,37–39,42,44,47,48,51,52,55,61–63, 66,70,75,78–81,83–85,88–90,92] or synchronous teleultrasound transmission [27,28,31,32,34–36, 41,43,49,50,53,54,59,60,64,65,67,74,76,77,82,86,87,91]. Several studies incorporated both transmission techniques [24,40,45,46,56–58,68,69,71–73,93]. The vast majority of studies transmitted data via internet, satellite, telephone or ISDN channels, however, three additional earlier studies incorporated physical courier transfer alongside internet transmission [58,66,68]. Internet transmission bandwidths ranged between 64 kilobits per second (Kb/s) to 9.6 gigabits per second (Gb/s) [36,42], however, many studies failed to provide such details. Included studies encompassed various

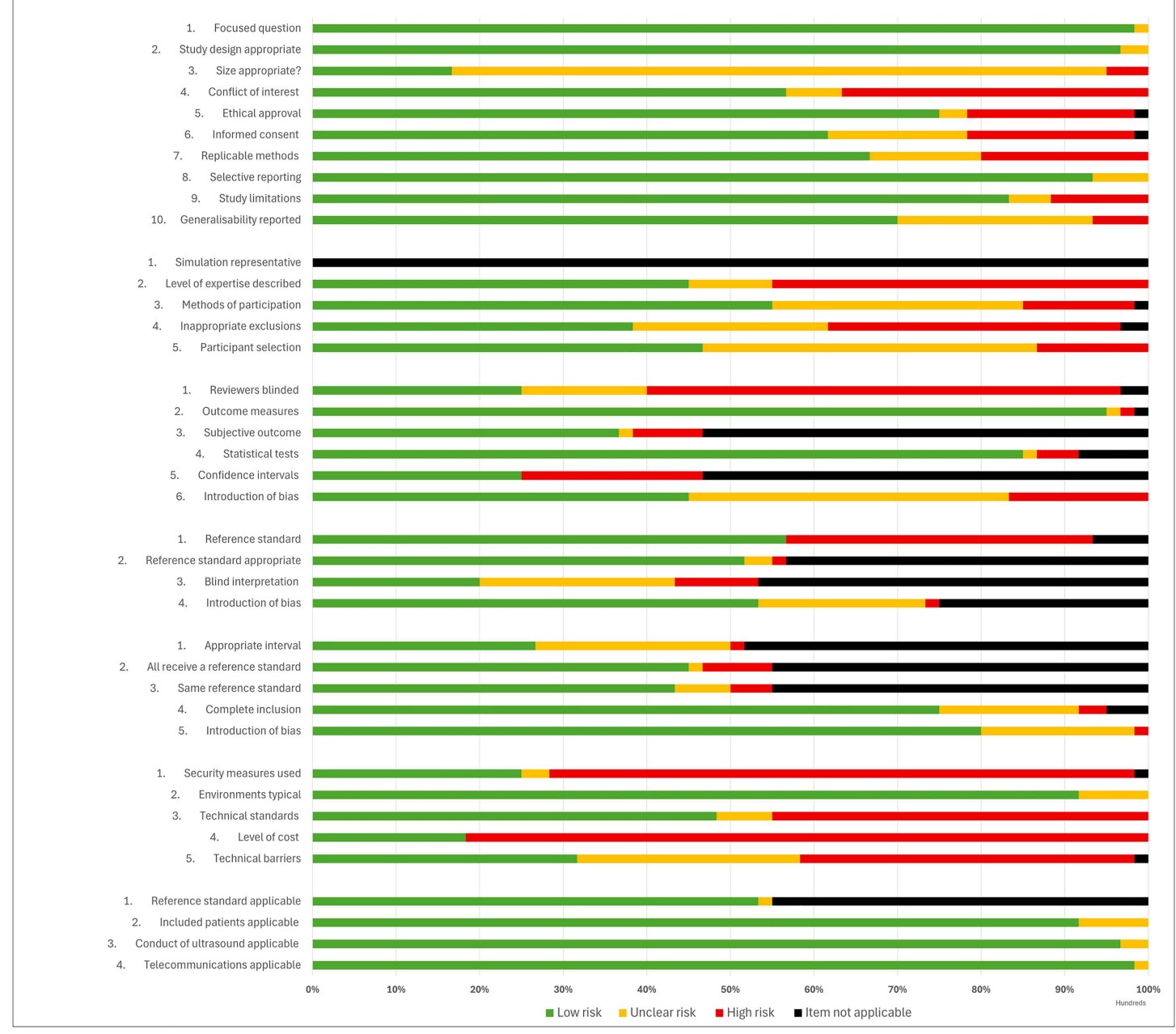

**Fig 4. Risk of bias results for clinical observational studies using the QUADAS-2 checklist, stratified by each domain.**

clinical applications including antenatal screening [34,35,38–41,45,46,57,58,63,68,69,74–76,79], antenatal diagnosis [25,29,30,32,33,42,47,48,51,55,56,60–62,64,80,82–84,86,88,90], access to specialist expertise in rural and deprived areas [23,24,26–28,31,36,43,49,53,54,59,66,72,77,85,91], and handheld/point-of-case ultrasound [37,44,50,52,65,67,70,75,81,87,89,92,93]. Several studies acknowledged more than one of these clinical applications. Documented distance between the operator and receiver for teleultrasound transmission, aside from in-hospital transfer ranged vastly from 4 to 8,276 miles [41,73]. Despite the crucial role of compression algorithms in facilitating the packaging and reduction of

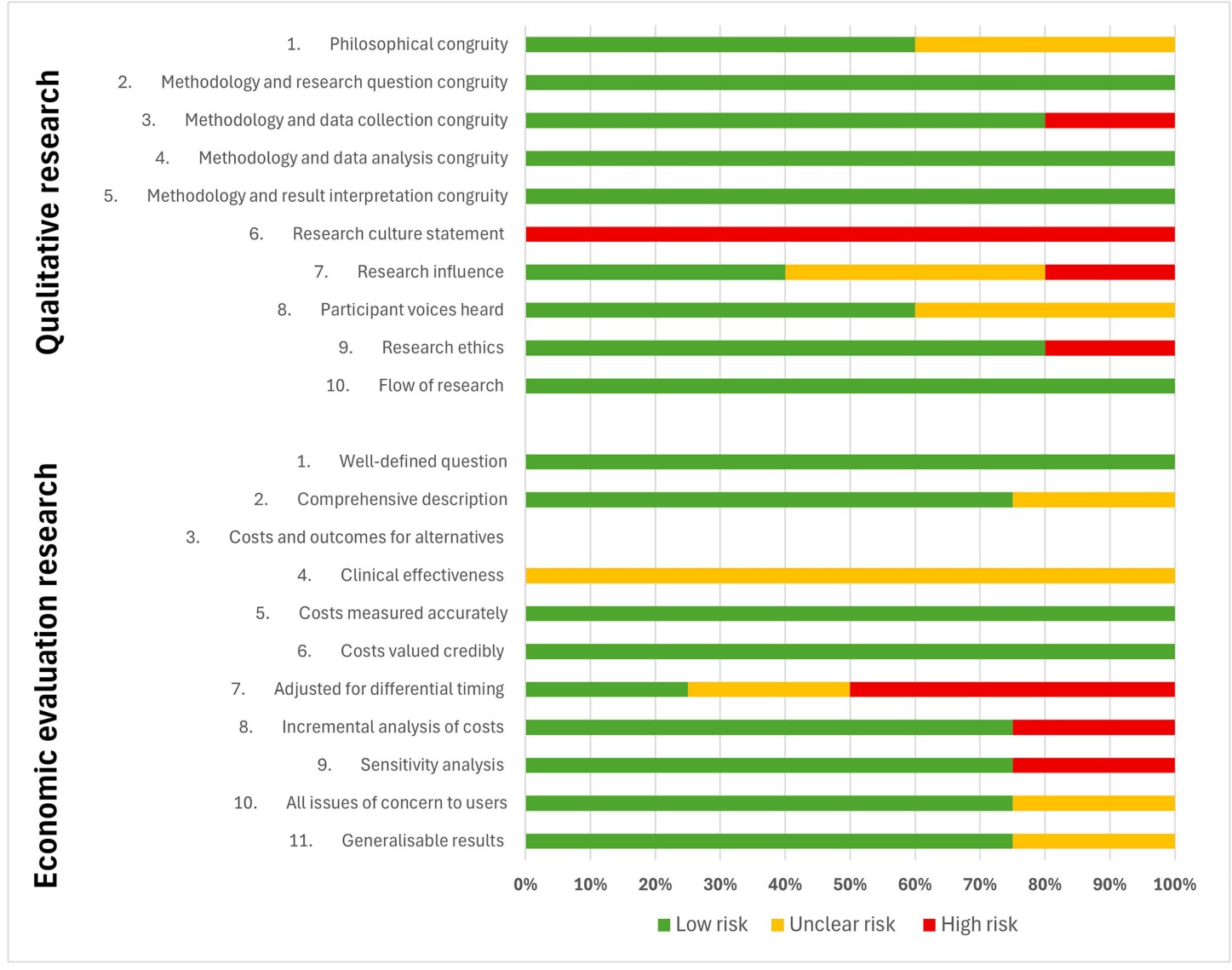

**Fig 5. Risk of bias results for qualitative and economic evaluating studies using the JBI frameworks, stratified by each domain.**

data volume from an original file format to enable successful teleultrasound, only eight studies detailed such information [40,41,43,57,59,63,84,88]. Detailed description of each study methodology can be found in S6 Table.

## Study findings

**Feasibility.** Thirty-three studies primarily evaluated feasibility outcomes. The duration of an ultrasound scan ranged from 3.8 min for a basic first-trimester scan to 38.1 min using a tele-operated robotic ultrasound device performing a second-trimester anatomy scan [24,68]. Frequently, if commented on, scans persisted longer than 10 min [28,35,40,57]. Transmission delays and failures were scarcely reported. In synchronous systems, transmission delays were commonly less than 3 s, with little impact on performance [24,28,40]. However, Adams and colleagues demonstrated ultrasound video transmission delays of up to 10 s in 14% of scans, whereby this impacted on performance were noted [24].

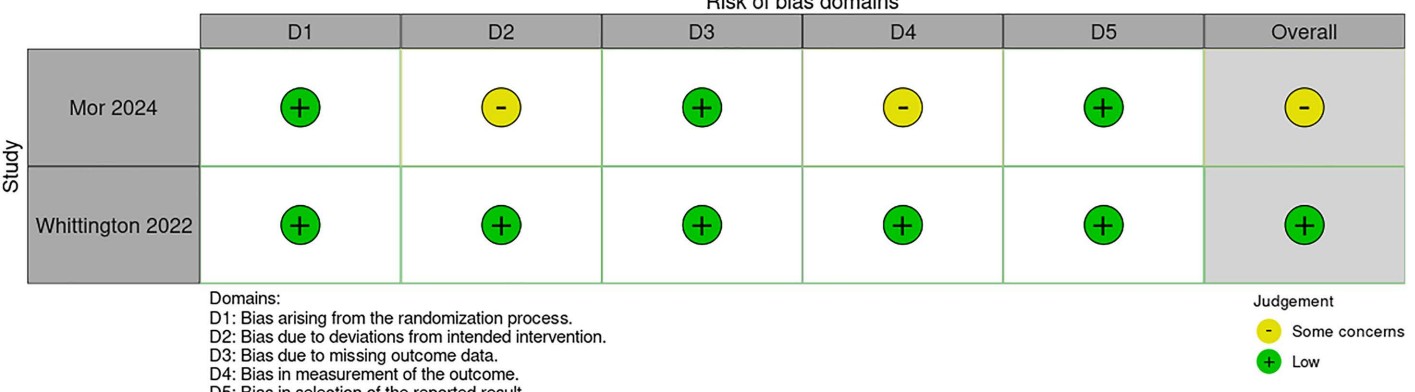

**Fig 6. Risk of bias results for randomized controlled trials using the Cochrane RoB2 tool, stratified by each domain.**

Amongst asynchronous systems, transmission delays ranged from 4 to 19 min; usually dependent on the number of image acquisitions acquired [26,27,81,84]. Transmission failure rates ranged from 0%–20%, however, minimal studies commented on this [27,67,68,74,77,86]. Primarily, this was attributed to insufficiencies in technological infrastructure support, equipment malformations, and software corruption.

Image quality was assessed using a variety of measures including binary assessment [26,57,63], Likert scales [41,66,68,74,84], and subjective metrics [24,28,40,43,45,86,89]. When studies compared against the reference standard, image quality of tele-transmitted scans was particularly good, frequently reporting more than 75% of tele-scans having similar quality or rather reporting high image quality agreement through subjective assessment [41,45,57,68,74]. Overall image quality was higher in 2-dimensional versus three-dimensional teleultrasound, primarily due to the loss of resolution on extracted planes from acquired volume data [66].

First- and third-trimester tele-scans frequently demonstrated high visualization scoring for basic anatomical identification, reporting rates ranging between 68%–100% and 81%–100%, respectively [24,26,27,40,44,53,59,67,68,78–80,89,93]. Studies examining second-trimester fetuses adequately identified structures ranging from 18% to 100% [23,24,47,53,57,59,60,63,70,80,88,93]. However, typically second-trimester scans examined a greater number of more specialized anatomical features. Included studies frequently classed an anatomical structure as correctly identified when the acquired images would be comparative to what is normally conducted in routine ultrasound practice. Table 2 represents a condensed tabulated summary of all visualization rates using teleultrasound. These have been stratified by anatomical structure and grouped by body system. A series of proportional meta-analyses demonstrating the identification rates for fetal/placental structures using teleultrasound, provided by included studies can be seen in S1 Fig. Generally, neurological, facial, and cardiac anatomy were frequently visualized at lower rates during routine anatomical teleultrasound surveys versus other structures. Such that the cavum septum pellucidum, nuchal thickness, fetal profile and genitalia were seen in only 33%, 43%, 64% and 80% of teleultrasound scans, respectively (S1 Fig). Fetal breathing was also infrequently identified in one study using patient-operated remote ultrasound at a rate of 24% [44]. Field of view, focal zone range, and gray scale changes following tele-transmission of scans were attributing factors to loss of image quality [40]. Equipment alterations, such as larger video monitors did enable enhanced clinician-reported image quality [43].

Meta-analysis of identification rates between teleultrasound and the reference standard were available in four studies for a subset of fetal and placental structures (Fig 7). Only two or three studies were eligible for each meta-analyzed outcome. Teleultrasound was non-inferior in 18 out of the 20 anatomical structures assessed when compared to the reference standard, with an overall pooled effect of RR 1.02 (95% CI [1.00,1.03]). Identification rates of the abdominal cord

**Table 2. Rates of identification reported within included studies for each respective anatomical structure.**

| Indication for teleultrasound | Rate of identification (cited studies) |
|---|---|
| **General** | |
| Amniotic fluid volume | 92.2%–100% [26,27,44,57,67,70,78,89,93] |
| Crown-rump length | 83% [68] |
| Fetal breathing | 23.8% [44] |
| Fetal movements | 83%–88.3% [44,89] |
| Fetal number | 83%–100% [57,68] |
| Fetal presentation | 100% [27,57] |
| Overall biometric assessment | 93.1% [27] |
| **Placenta** | |
| Cord vessel number | 100% [57] |
| Placental location | 93.1%–100% [26,27,57] |
| Placental cord insertion | 85.5% [57] |
| **Neurological** | |
| Cavum septum pellucidum | 33% [23] |
| Cerebellum | 78.5%–90% [23,57] |
| Choroid plexus | 90% [23] |
| Cisterna magna | 78.5%–97.1% [23,54,57] |
| Cranium | 87.5%–100% [23,26] |
| Midline falx | 90% [23] |
| Nuchal thickness | 43.5% [57] |
| Ventricles | 88.9%–100% [23,57,78] |
| **Facial** | |
| Face/lips | 50%–93% [23,53,57] |
| Nasal bones | 100% [78] |
| Orbits | 90% [23] |
| Profile | 17.9%–94.3% [53,57,70] |
| **Skeletal** | |
| Chest | 62.5% [23] |
| Femur | 87.5% [26] |
| Spine | 55.6%–100% [23,53,57,78] |
| **Cardiac** | |
| Aortic and pulmonary valves | 90%–100% [60,80] |
| Aortic arch | 76.7%–100% [57,60,63] |
| Atrio-ventricular valves | 98%–98.6% [60,80] |
| Axis | 80%–96% [23,80] |
| Cardiac Doppler | 74%–100% [68,78,93] |
| Cardiac chambers | 71.4%–100% [23,47,57,60,63,73,79,80,88] |
| Crux | 94%–96.7% [63,80] |
| Crossing over of great arteries | 86%–100% [60,80] |
| Ductal arch | 100% [60] |
| Fetal heart activity | 52.2%–98.7% [44,70,89,93] |
| Foramen ovale | 100% [80] |
| Inferior/superior vena cava | 83.3% [63] |
| Left ventricular outflow tract | 40%–100% [23,47,53,57,60,63,80,88] |
| Primary atrial septum | 98.6% [60] |
| Pulmonary venous connections | 83.3%–98% [60,63,80] |

*(Continued)*

**Table 2.** (Continued)

| Indication for teleultrasound | Rate of identification (cited studies) |
|---|---|
| Right ventricular outflow tract | 40%–100% [23,47,53,57,60,63,80,88] |
| Situs | 70%–100% [23,60,63] |
| Three-vessel view | 92%–100% [47,63,80,88] |
| Three vessels and trachea view | 74%–92% [47,80] |
| Ventricular septum | 84.1% [60] |
| **Gastric** | |
| Abdomen | 100% [23,26,57,88] |
| Cord insertion | 99%–100% [23,57] |
| **Renal** | |
| Bladder | 100% [23,57] |
| Kidneys | 55.6%–100% [23,53,57] |
| **Reproductive** | |
| Genitalia | 68.6%–88.5% [53,57] |
| **Extremities** | |
| Arm | 95%–100% [23,57] |
| Foot | 97.5%–100% [23,57] |
| Hand | 74.3%–100% [23,53,57] |
| Leg | 99%–100% [23,57] |

insertion and the fetal hand were significantly greater in the teleultrasound group RR 1.09 (95% CI [1.04,1.14]) and RR 1.06 (95% CI [1.01,1.12]), respectively. The reference standard in the included studies was not consistent, ranging from conventional in-hospital ultrasound to videotape review. This likely attributed to variations in the point estimates seen and thus should be considered when interpreting these findings.

The relationship between various bandwidth requirements and image quality was further explored within studies. Very low bandwidths (<284 Kb/s) demonstrated significantly worse ($p < 0.01$) image quality versus higher bandwidths; however, the most cost-effective rates were 384 Kb/s and 1 megabit per second (Mb/s) [35,74]. Decreasing the frame rate may reduce overall data burden, enabling transmission over lower bandwidths, whilst aiming to not decrease overall image quality and risk transmission failure [57,77]. Bandwidths above 2Mb/s alongside a higher framerate demonstrated ultrasound images which were comparative to original scans, with minimal loss of image quality [40,43,45,50]. However, certain studies still demonstrated constraints with image quality and delayed video lag using advanced telesonography techniques with bandwidths up to 5Mb/s, such as robotic ultrasound [24,28]. Less recent publications (>20 years old) frequently used bandwidths of 384 Kb/s or less [34,57,68,74,77], whilst more recent studies, with the advent of newer teleultrasound technology all used bandwidths ≥1Mb/s [24,26,28,40,41,45,50]. Lower bandwidth requirements were more frequently seen in studies with a primary clinical focus on antenatal screening, whilst higher bandwidth requirements were more commonly seen in studies further aiming to improve access to specialist expertise in remote and deprived areas. It was feasible to transmit ultrasound data across a wide range of distances, both asynchronously and synchronously, with many studies transmitting data over 500 kilometers (km) [24,28,34,35,38,41,66,77]. Evidence suggests that healthcare professionals with novice obstetric ultrasound skills are capable of performing basic scans with the aid of long-distance remote tele-supervision [26,28,50,81]. A summary of studies primarily evaluating feasibility outcomes can be seen in Table 3.

**Diagnostic accuracy.** Twenty-seven studies primarily evaluated outcomes in relation to the diagnostic accuracy of teleultrasound. For almost all such studies, accuracy was established by the comparative proportion of correctly classified cases using teleultrasound versus the reference standard. Most studies used either conventional in-hospital ultrasound

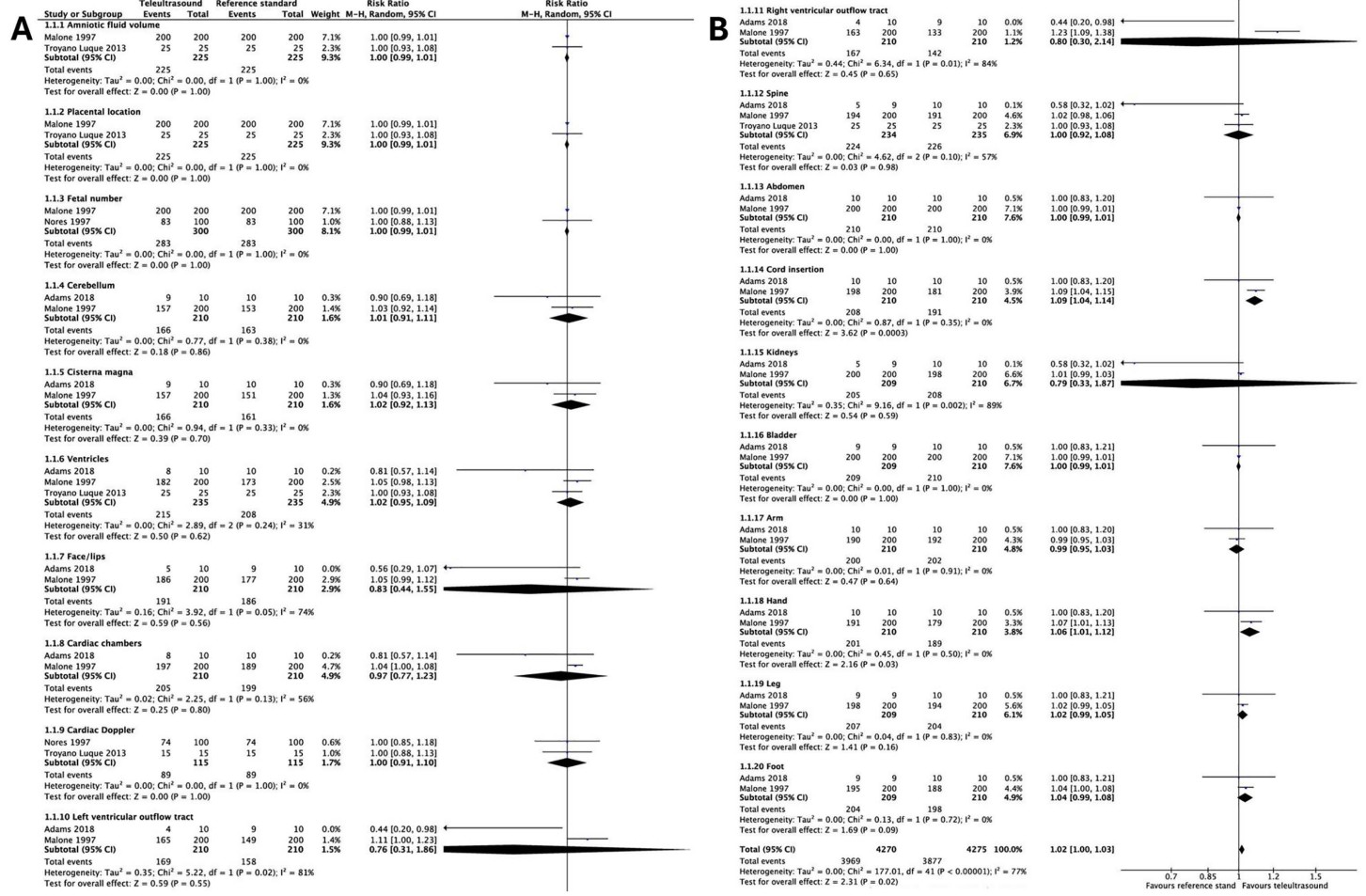

**Fig 7. Meta-analysis of identification rates for fetal and placental structures using teleultrasound vs. the reference standard. (A)** Structures 1–10. **(B)** Structures 11–20, including the overall pooled effect. CI, confidence interval. M-H, Mantel–Haenszel.

[23,27,37,42,47,51,53,60,78,79,82–84,88,91,93], intraoperative diagnosis [90] or postnatal diagnosis as the chosen reference standard [25,33,36,55,56,82]. Some studies graded each teleultrasound scan employing a binary classification (normal/abnormal or satisfactory/unsatisfactory) [29,44,52]. However, trinary and quinary classifications were implemented in certain studies, to further refine the diagnostic accuracy of teleultrasound when comparing to the reference standard [25,42,53,79]. A single study, by Rabie and colleagues explicitly defined accuracy as the proportion of correctly identified cases by the sum of the true positives and true negatives [71].

Diagnostic accuracy ranged between 68.7% and 98.6% [82,83], with studies frequently citing above 95% [32,47,55,56,60,82]. One randomized controlled trial of 300 patients, whereby a sonographer performed the scan with synchronous interpretation by a group of five specialists, demonstrated that teleultrasound was non-inferior to conventional ultrasound for assessing congenital anomalies. Diagnostic accuracy was above 97% for both groups, however, details regarding the type of anomalies detected and their respective individualized diagnostic accuracy were not reported [82].

A HSROC model was applied to evaluate the diagnostic accuracy of teleultrasound across available included studies (Fig 8). The pooled diagnostic accuracy across studies demonstrated excellent overall performance, with an AUC of 0.93,

**Table 3. Summary of studies primarily evaluating feasibility outcomes.**

| Study (year) | Sample size | Recruitment location | Communication and transmission | Summary of results |
|---|---|---|---|---|
| Adams (2020) [24] | 18 | Canada | • Internet<br>• Synchronous and asynchronous | 81% of limited obstetric exams (9/11) were adequate for subjective interpretation. 20% of the second-trimester exams (2/10) were adequate for subjective interpretation. Transmission delays of up to 10 seconds are noted in 14% of scans. |
| Ahmed (2024) [86] | 513 | Egypt and one other country (not specified) | • Internet<br>• Synchronous | Tele-echocardiography and counseling were successful in all the cases. Satisfaction with the service was 3.8/4, with the main limitation being the need for further referral to a tertiary center for delivery. |
| Arbeille (2014) [26] | 8 | Romania, French Guyana. Spain | • Internet<br>• Asynchronous | Organs scanned in the obstetrical cases were adequately visualized by the expert in seven of eight (88%) examinations of the fetal head, femur, and umbilical cord and eight of eight (100%) examinations of the fetal abdomen and placenta. Transmission delay and processing ranged from four to 19 min. |
| Arbeille (2005) [27] | 29 | Spain | • Telephone or satellite<br>• Synchronous | In 93.1%, all biometric parameters, placental location, and amniotic fluid volume, were correctly assessed. Transmission delay was 1 s and transmission failures occurred in approximately 9% of scans. |
| Arbeille (2016) [28] | 15 | France | • Internet<br>• Synchronous | Image qualitative subjectively reported as similar to conventional ultrasound images. Transmission delay of three seconds. Image quality not affected using internet connections in different settings. |
| Axelrod (2025) [87] | 20 | Israel | • Internet<br>• Synchronous | Remote visits had a success rate of 97.4% (38 of 39), with significantly shorter durations compared with in-clinic visits (median 59.0 min vs. 159.0 min, $P < 0.001$). Women expressed high satisfaction (6.6 of 7), and adherence to the hybrid model. |
| Chan (2000) [34] | 24 | Australia | • Internet<br>• Synchronous | Modifications to clinical diagnosis were made in 45.8% of cases and modification to the management plan in 33.3% of cases. Specialists and referring clinicians rating their confidence and usefulness of telesonography highly on Likert scale. |
| Chan (2001) [35] | 71 | Australia | • Internet<br>• Synchronous | Overall, the consultations resulted in some modifications to the clinical diagnosis in 41% of the cases, and modifications to the management plan in 40% of the cases. Clinicians and patients rated the teleconsultations highly. |
| El Guindi (2013) [39] | 33 | French Guyana | • Internet<br>• Asynchronous | Isolated cardiovascular malformations were detected in 23 fetuses. Extracardiac abnormalities were identified in 8 fetuses. Offline analysis of cardiovascular anomalies had diagnostic advantages over two-dimensional ultrasound. |
| Ferlin (2012) [40] | 20 | Brazil | • Internet<br>• Synchronous and asynchronous | All fetal structures could be viewed and the quality of images received by the examiners was considered subjectively normal. Transmission delays were one to three seconds. The use of low-cost teleultrasound software appeared suitable for screening for chromosomal abnormalities in the first trimester of pregnancy. |
| Ferreira (2014) [41] | 15 | Brazil | • Internet<br>• Synchronous | The quality of the original video clips was slightly better than that observed by the transmitted video clips by Likert scale. In 47/60 cases (78.3%), the quality of the video clips were judged to be the same. In 12/60 (18.3%), the original video clips were judged as better quality compared to the tele-transmitted clips. |
| Fisk (1995) [43] | 6 | United Kingdom | • Telephone<br>• Synchronous | As a result of teleultrasound, five patients were spared travel. Consultants found themselves confidently making diagnosis and carrying out counseling. However, audio transmission issues arose at the initial phase of the study, leading to technological challenges. |
| Hishitani (2014) [45] | 117 | Japan | • Telephone or internet<br>• Synchronous and asynchronous | The number of emergent transportations of neonates with severe cardiac anomalies continued to drop. Telediagnosis was requested even for cases of mild abnormalities, thus the number of false positives increased. Larger video monitors did enable enhanced clinician-reported image quality |
| Inamura (2021) [48] | 2 | Japan | • Internet<br>• Asynchronous | Using spatio-temporal image correlation (STIC), the left aortic arch was visualized. The combined use of color imaging with STIC for remote diagnosis enables the diagnosis of many congenital heart diseases. |
| Ishikawa (2023) [49] | 16 | Japan | • Internet<br>• Synchronous | 90.0%, 80.0%, and 70.0% of women perceived improvements in their physical, mental, and economic burdens, respectively. Although 70.0% of participants experienced anxiety before the introduction, all were satisfied after delivery. |

*(Continued)*

| Study (year) | Sample size | Recruitment location | Communication and transmission | Summary of results |
|---|---|---|---|---|
| Jemal (2024) [50] | NR (100 scans) | Ethiopia | • Internet<br>• Synchronous | Concordance between healthcare providers' and obstetricians' image interpretations ranged from 79% to 100% for each parameter assessed. 99.4% of participants surveyed indicated that they would recommend antenatal ultrasound using tele-ultrasound to friends and family |
| Leighton (2019) [54] | 6,757 | USA | • Internet<br>• Synchronous | Telemedicine patients experienced similar outcomes to the in-person group, indicating that telemedicine can serve as an effective substitute for in-person care. An overwhelming majority of telemedicine patients were satisfied with their visit and indicated that they would be interested in receiving care via telemedicine in the future. |
| Le Vance [89] | 15 | United Kingdom | • Internet<br>• Asynchronous | Overall, the fetal heartbeat, movements, and an assessment of the liquor volume were identified in 92%, 83%, and 100% of all ultrasound scans, respectively. 79% of all scans had all three criteria unanimously assessed. |
| Malone (1997) [57] | 200 | USA | • Telephone<br>• Synchronous and asynchronous | Telemedicine and videotape interpretations provided similar scores in 84% of scans. In 17 of the 33 anatomic categories telemedicine provided significantly better scores ($p < 0.05$) than videotape, whereas in the remaining 16 anatomic categories the scores were equivalent. |
| Michailidis (2001) [63] | 30 | United Kingdom | • Internet<br>• Asynchronous | A complete heart examination was accomplished in 23 of 30 cases. Identification rates using binary assessment for cardiac anatomy ranged from 76.7% to 100%. |
| Nelson (2001) [66] | 56 | USA | • Internet and courier<br>• Asynchronous | Overall, three-dimensional ultrasonography could produce diagnostic-quality results comparable with those of two-dimensional ultrasonography using Likert scale. Three-dimensional ultrasonographic image quality was lower than that of two-dimensional ultrasonography. |
| Nir (2024) [67] | 10 | Israel | • Internet<br>• Synchronous | Nine women (90%) were able to complete remote modified biophysical profile assessment. Satisfactory amniotic fluid volume measurements were achieved in 100% of participants. The telemedicine encounter was significantly shorter (93.1 ± 33.1 min) than the in-person visit (247.2 ± 104.7 min) ($p < 0.001$). |
| Nores (1997) [68] | 100 | USA | • Telephone and courier<br>• Synchronous and asynchronous | Telemedicine and videotape review scores were the same in 95 out of 100 cases. No transmission failures occurred. In three cases, telemedicine had higher scores than videotape review using Likert scale. |
| Oelmeier (2023) [69] | 59 | Germany | • NR<br>• Synchronous and asynchronous | In 76% (53/70) of cases, the video consultations provided useful information and a knowledge gain for providers on both sides. Overall, 47% (33/70) of the in-house visits could be avoided through video consultations. |
| Pardo (2025) [92] | 107,167 | Israel | • Internet<br>• Asynchronous | Users had higher socioeconomic scores, were more primiparous and had a higher incidence of chronic disease and pregnancy complications. Preterm birth rates and adverse neonatal outcomes did not differ between groups. |
| Pontones (2023) [70] | 46 | Germany | • Internet<br>• Asynchronous | Success rates for locating the target structure were 52.2% for videos of the fetal heartbeat, 52.2% for videos of the amniotic fluid in all four quadrants and 17.9% for videos of the fetal profile. |
| Reddy (2000) [72] | 49 | Cananda | • Telephone<br>• Synchronous and asynchronous | The technical quality of the transmitted images was reported as excellent. The teleradiology system shortened the time it took for patients to be informed of their examination results (1 week on average for the control group vs. the same day for the teleradiology group). |
| Schwartz (2021) [73] | 122 | USA | • Internet<br>• Synchronous and asynchronous | All the fetal echocardiograms suspicious for congenital heart disease were confirmed on postnatal echocardiograms. To their knowledge, none of the normal fetal echocardiograms were found to have congenital heart disease postnatally. |
| Sharma (2003) [74] | 34 | USA | • Internet<br>• Synchronous | Surveys from patients with direct physician contact and by telemedicine showed high satisfaction with telemedicine-assisted screening and counseling. |
| Sheilds (2022) [75] | NR | USA | • Internet<br>• Asynchronous | A significant decrease in patient visits following the conversion to a telehealth platform (53.35 visits per day vs. 40.3 visits per day, $p < 0.001$). There was an increase in more basic follow-up ultrasound procedures, complexity over comprehensive follow-up ultrasound procedures after conversion. |

*(Continued)*

**Table 3.** (Continued)

| Study (year) | Sample size | Recruitment location | Communication and transmission | Summary of results |
|---|---|---|---|---|
| Soong (2002) [77] | 160 | Australia | • Internet<br>• Synchronous | Transmission failure occurred in 20% of tele-scans. 94% of the women strongly agreed or agreed when asked if their privacy and confidentiality were maintained during the videoconference. |
| Vinals (2005) [80] | 50 | Chile | • Internet<br>• Asynchronous | A telemedicine link via the internet was possible in all cases. A complete cardiac examination according to set criteria was achieved by the administrator in 86% of the cases scanned by one operator and 95% of the cases scanned by the other operator. |
| Vinayak (2018) [81] | NR (271 scans) | Kenya | • Internet<br>• Asynchronous | The average turn-around time for post scan validation was 12 min of which 4 min was required for transmission of scans to the base hospital. No problems were reported in respect to the mobile phone, modem, or teleradiology system. |

NR, not reported; USA, United States of America.

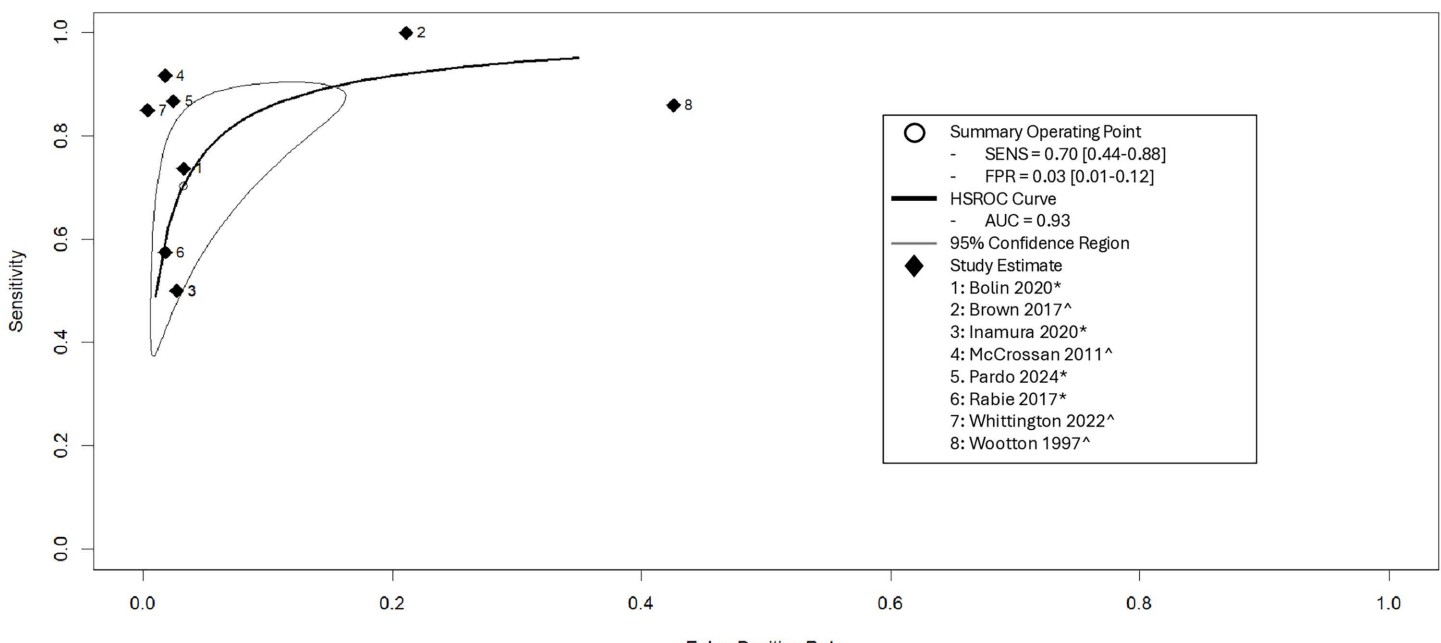

**Fig 8. Hierarchical summary receiver operating characteristic (HSROC) curve for the diagnostic accuracy of teleultrasound with the confidence region around summary operating point.** (*) represents the reference standard was conventional in-hospital ultrasound. (^) represented the reference standard was postnatal diagnosis. AUC, area under the curve; FPR, false positive rate; HSROC, hierarchical summary receiver operating characteristic SENS, sensitivity.

indicating strong discriminative ability. The pooled sensitivity was 0.70 (95% CI [0.544,0.88]), and the pooled false positive rate was 0.03 (95% CI [0.01,0.12]). Statistical heterogeneity was moderate, with an I² value of 42.5%. This level of heterogeneity is likely indicated by some variability in study populations and choice of reference standard. Additionally, a proportional meta-analysis of all available accuracy results within included studies can be seen in S2 Fig. Overall, the pooled diagnostic accuracy of obstetric teleultrasound was 93% (95% CI [87%,99%]). A tabulated version including sensitivity, specificity, positive and negative predictive values can be seen in S7 Table.

Among users reviewing obstetric tele-transmitted images/videos, inter-agreement was frequently moderate to excellent, albeit there was wide range; Cohen's kappa ($k$) = 0.11–1.00 [37], $k$ = 0.75 [42], $k$ = 0.70–0.98 [44], $k$ = 0.17–0.65 [53], $k$ = 0.59 [56], $k$ = 0.89 [60], $k$ = 0.91 [78], $k$ = 0.40–1.00 [79], $k$ = 0.55–0.78 [84], $k$ = 0.94–1.00 [85], $k$ = 0.63 [90]. Low agreement levels were primarily in relation to discriminate positions of placental locations and identification of more subtle anatomical features such as the fetal kidneys, which frequently rely on optimal image quality for clear identification [37,53]. Intraclass correlations between teleultrasound and conventional ultrasound were noted by three studies, demonstrating poor to excellent agreement 0.711–0.995 [26], 0.28–0.95 [84] and 0.990–0.993 [91]. A summary of studies primarily evaluating diagnostic accuracy outcomes can be seen in Table 4.

**Clinical utility.** Maternal-fetal clinical outcomes were not significantly different ($p > 0.05$) between the teleultrasound group versus the control groups; however, rates of prenatal diagnosis significantly increased following implementation of teleultrasound, ranging from 26% to 159% ($p < 0.05$) [29,54,62,93]. Regarding system-based findings, teleultrasound shortened the overall clinic visit time, reduced the time taken for patients to be informed of clinical findings, and was associated high ratings of acceptability among service users [54,72,87]. Teleultrasound was noted to decrease patient waiting times for scans and reduced overall patient backlogs, fostering a streamlined obstetric clinic [59]. Chan and colleagues revealed that the implementation of teleultrasound resulted in a modification to the existing clinical diagnosis in 46% of teleultrasound cases, which altered the continued management plan in a third of cases (half of which were minor variations) [34].

**Patient-operated telesonography.** Eight studies assessed the usage of patient-operated obstetric teleultrasound [44,65,67,70,87,89,92,93]. Patient-operated ultrasound was a feasible solution for performing a basic assessment of fetal well-being, with high user acceptability. Hadar and colleagues and Le Vance and colleagues demonstrated high acquisitions of the fetal heartbeat (95.3% and 92%), fetal movements (88.3% and 83%), and assessment of normal liquor volume (92.2% and 100%) in 1,360 and 24 self-scans, respectively [44,89]. Pontones and colleagues reported lower identification rates using asynchronous transmission in 46 women, at only 52.2% for the fetal heart rate and amniotic fluid volume, whilst only 14% for fetal profile [70]. Contrastingly, Nir and colleagues incorporated real-time support for 10 patient self-scan sessions, demonstrating appropriate biophysical assessment in 90% of cases, combined with high acceptability ratings [67]. Further comparison between patient independently scanning versus those synchronously guided by clinician demonstrated improved accuracy across several parameters, including the fetal heart rate, measurement of the liquor volume using the latter technique [93]. The use of patient-operated devices has also demonstrated effectiveness for anxiety management for women with recurrent pregnancy loss, as demonstrated in a recently published randomized controlled trial of 50 women [65]. Patient-operated teleultrasound technology is a novel intervention and currently only provides limited assessments of fetal well-being. Further evaluation of home use is necessitated, particularly in patients from ethnic minorities and low education levels, which were underrepresented in current studies [44,65,89]. A summary of studies using patient-operated teleultrasound devices can be seen in Table 5.

**Three- and four-dimensional ultrasound.** Three-dimensional obstetric ultrasound via asynchronous tele-transmission was investigated by five studies [26,42,55,63,66]. Primarily, image quality was poor in comparison to two-dimensional ultrasound, with far longer image pre-processing (up to 4 hours) and transmission times due to large data volumes [42,55,66]. Nevertheless, Mabuchi and colleagues reported a high prenatal diagnostic accuracy (95%) for cases of congenital heart disease in second and third trimester fetuses, whilst Ferrer-Roca and colleagues demonstrated a moderate interclass correlation (Kappa = 0.70) between three-dimensional and conventional image acquisitions within all trimesters [42,55]. This scan modality may play a role in off-line assessment for remote locations whereby access to a tertiary unit is distant [63].

Four-dimensional image acquisition demonstrated a promising ability to adequately recognize fetal cardiac structures within all gestational ages via asynchronous transmission [25,79,80]. Vinals and colleagues exhibited a moderate interobserver concordance (Kappa > 0.6) within 14 out of 18 intracardiac structures assessed within the first trimester [79].

**Table 4. Summary of studies primarily evaluating diagnostic accuracy outcomes.**

| Study (year) | Sample size | Recruitment location | Communication and transmission | Summary of results |
|---|---|---|---|---|
| Adams (2018) [23] | 30 | Canada | • Internet<br>• Asynchronous | Intraclass correlations showed excellent agreement (>0.90) between telerobotic and conventional measurements of all 4 biometric parameters. Of 21 fetal structures included in the anatomic survey, 80% of the structures attempted across all patients were sufficiently visualized by the telerobotic system (range, 57% to 100% per patient). |
| Adriaanse (2012) [25] | 10 | Netherlands | • NR<br>• Asynchronous | Ten second-trimester spatiotemporal image correlation (STIC) volumes were sent to three observers in different tertiary care centers with expertise in 4D echocardiography. In two cases, all observers correctly diagnosed all details of the volume datasets. The observer with the best performance reached perfect agreement in six cases and nearly perfect agreement in three. |
| Beldjerd (2022) [29] | 260 | France | • Internet<br>• Asynchronous | In our series, asynchronous analysis allowed the required physician to make an accurate diagnosis and identify 74 (28.5%) pregnancies associated with malformations and rule out abnormalities in 186 (71.5%) of the cases. The asynchronous tele-expertise was not associated with face-to-face consultations for 72.7% (189/260) of the patients, who without moving, were able to have access to a precise diagnosis by ruling out the presence of anomalies in 163/189 of these patients and confirming them in 26/189 patients. |
| Bolin (2020) [32] | 1,164 | USA | • Internet<br>• Synchronous | In 1,164 fetuses, fetal tele-echocardiography identified all types of congenital heart disease, with a sensitivity of 74% and specificity of 97%. For the detection of ductal-dependent congenital heart disease, fetal tele-echocardiography was 100% sensitive and specific. Between 2009 and 2018, annual statewide prenatal detection rates of congenital heart disease requiring heart surgery in the first 6 months of life rose by 159% ($p < 0.01$). |
| Brown (2017) [33] | 75 | USA | • Internet<br>• Asynchronous | For identifying complex congenital heart disease, fetal tele-echo had a sensitivity and specificity of 100%. The average number of fetal echocardiograms per mother– infant pair was 1.1. |
| Cuneo (2019) [36] | 369 | USA | • Internet<br>• Asynchronous | Four hundred and fifty-five telecardiology encounters. Congenital heart disease or arrhythmia was diagnosed in 28 and 15 fetuses, respectively; there was one false-negative result. All fetuses were correctly risk-stratified with respect to delivery location. |
| Day (2025) [88] | 48 | United Kingdom | • Internet<br>• Asynchronous | The addition of second review improved the sensitivity to 0.975 using video clips, which was significantly higher than using still images (0.892, $p = 0.002$). There was a significant drop in specificity to 0.767 and 0.833 ($p < 0.001$) for the video and still method, respectively, which were statistically similar to each other ($p = 0.117$). |
| Dougherty (2021) [37] | 91 | USA | • NR<br>• Asynchronous | The agreements among readers and between teleultrasound readers and the gold standard ultrasound report for the placental location, fetal number, fetal presentation and biometric parameters ranged from 0.11–1.00. |
| Ferrer-Roca (2006) [42] | 32 | Spain | • Internet and ISDN<br>• Asynchronous | Good correlation ($k = 0.7$) was found between local and distant diagnoses using three-dimensional volumetric reconstruction. In 30%, images were considered of low quality and in 29% of good quality; diagnosis could be done with confidence in all except 7 cases. |
| Hadar (2022) [44] | 100 | Israel | • NR<br>• Asynchronous | Interobserver agreement was 94.4% for fetal heart rate activity, 85.9% for body movements, 69.5% for fetal tone, 86.9% for amniotic fluid volume, and 94.0% for breathing movements for home ultrasound with tele-transmission functionality. |
| Inamura (2020) [47] | NR | Japan | • Internet<br>• Asynchronous | The sensitivity and specificity of spatio-temporal image correlation (STIC) in congenital heart disease screening was 50% and 99.5%, respectively. The sensitivity and specificity of STIC in screening for severe congenital heart disease was 82% and 99.9%, respectively. |
| Kern-Goldberger (2021) [51] | NR | USA | • Internet<br>• Asynchronous | The overall rate of the potential missed diagnoses was 34.5% and varied significantly by type of ultrasound (anatomy scans vs. other first-, second-, and third-trimester ultrasounds) ($p < 0.01$). Moreover, there were significant differences in the rate of the potential missed diagnoses by organ system, with the highest rate for cardiac anomalies ($p < 0.01$). |

*(Continued)*

**Table 4.** (Continued)

| Study (year) | Sample size | Recruitment location | Communication and transmission | Summary of results |
|---|---|---|---|---|
| Kozuki (2016) [84] | 804 | Nepal | • Internet<br>• Asynchronous | Each auxiliary nurse–midwife's k statistic for diagnosis of non-cephalic presentation using home portable ultrasound was above 0.90 compared with the remote ultrasonogram reviewers. For multiple gestation, the auxiliary nurse–midwives were in perfect agreement with both the ultrasonogram reviewers. |
| Landwehr (1997) [53] | 35 | USA | • ISDN<br>• Synchronous | There was complete consistency of interpretation for 25 of 38 (66%) fetal structures using teleultrasound. Agreement levels ranged from 69% to 97% (k=0.17, 0.65). |
| Mabuchi (2020) [55] | 182 | Japan | • Internet<br>• Asynchronous | Congenital heart disease (CHD) using teleultrasound was detected in 14.9% of cases (24/161); the accuracy of prenatal diagnosis was 95.0% (153/161). The ultrasonographic telediagnosis system identified seven severe cases with CHD requiring immediate postnatal surgical or medical treatment. |
| Manley (2024) [59] | 460 | USA | • Satellite<br>• Synchronous | TeleScan provided visualization of critical anatomical structures in both the first and second trimesters with 100% reliability. Fetal sonography studies performed in the late third trimester are often limited due to acoustic attenuation and fetal positioning; however, the service provided visualization of critical structures with greater than 90% reliability. |
| McCrossan (2011) [60] | 67 | United Kingdom | • ISDN<br>• Synchronous | 69 remote fetal echocardiograms were performed and showed 58 normal hearts and 11 with congenital heart disease. Telemedicine with live guidance was accurate in 97% of cases compared with local assessment at the regional center ($k=0.89$) indicating excellent agreement. |
| Meiman (2022) [62] | 1,270 | USA | • Internet<br>• Asynchronous | The rate of prenatal diagnosis prior to the implementation of the first fetal tele-echocardiography site was 13.8% and after the sites were established, the prenatal diagnosis rate was 39.7% ($p<0.01$). |
| Nieto-Calvache [90] | 60 | Argentina, Brazil, Colombia, Egypt, England, Ghana, Indonesia, Ireland, Italy, Taiwan, USA | • Internet<br>• Asynchronous | Teleconsultant antenatal evaluation and management plans matched those of the local team in 71.7% of the cases. The severity was overestimated in nine reviews (16.9%) and was underestimated in six reviews (11.3%). Using the Fleiss' Kappa co-efficient gave a value of 0.63, indicating substantial or considerable agreement. |
| Olsen (2025) [91] | 46 | Norway | • Internet<br>• Synchronous | Biometric measurements showed excellent reliability (intraclass correlation coefficient 0.990–0.993) with acceptable limits of agreement for robotic ultrasound. Twenty questions about patient experiences were asked and 94% of the women scored for highest level of satisfaction. |
| Pardo (2024) [93] | 28 | Israel | • Internet<br>• Synchronous and asynchronous | Rates of successful utilization of the Pulsenmore tool for measurement of the fetal heart rate were 84.7±11.24% of scans made in app-guided mode and 96.3±6.35% of scans made in clinician-guided mode. Corresponding values for the maximum vertical pool were 91.7±2.31% and 95.0±1.73%. The sensitivity for evaluating maximum pool depth was 87.5% and 100% in app-guided and clinician-guided modes. Specificity was 95% and 95.5% in both modes, respectively. |
| Rabie (2017) [56] | 2,368 | USA | • Internet<br>• Synchronous and asynchronous | The teleultrasound cohort had a congenital anomaly prevalence of 5.66%. The sensitivity of teleultrasound was 57.46%; the specificity was 98.21%; and the accuracy was 95.9%. Anatomic surveys were completed after 1 visit in 82% of patients, whereas 63% and 61% of the remaining patients required 2 and 3 visits, respectively. |
| Rabie (2019) [71] | 5,903 | USA | • Internet<br>• Synchronous and asynchronous | The sensitivity of teleultrasound and on-site ultrasound was 57.46% and 76.57%, and the accuracy was 95.9% and 90.97%, respectively. The accuracy, specificity, positive, and negative predictive values of teleultrasound are similar to on-site ultrasound. |
| Toscano (2021) [84] | 126 | Peru | • Internet<br>• Asynchronous | Obstetric ultrasound telediagnosis showed excellent agreement with standard of care ultrasound allowing the identification of number of fetuses (100% agreement), fetal presentation ($\kappa=0.78$), placental location ($\kappa=0.74$), and assessment of normal/abnormal amniotic fluid volume (99.2% agreement). Intraclass correlation was poor to excellent for all fetal biometric measurements (0.28–0.95). |

*(Continued)*

**Table 4.** (Continued)

| Study (year) | Sample size | Recruitment location | Communication and transmission | Summary of results |
|---|---|---|---|---|
| Troya-noLuque (2013) [78] | 25 | Spain | • Internet<br>• Asynchronous | Ultrasound examination was performed transvaginal with tele-transmission. Pearson's correlation coefficient was high for biparietal diameter (0.72) and for overall correlation (0.997). |
| Vinals (2008) [79] | 49 | Peru | • Internet<br>• Asynchronous | Concordance of visualizing different structures and views of the fetal heart using spatiotemporal image correlation telemedicine by two reviewers ranged from 0.397 to 1.00. |
| Wootton (1997) [83] | 49 | United Kingdom and Ireland | • Internet<br>• Asynchronous | Of the 84 recordings transmitted at 1920kbit/s, 71 (85%) were diagnosed correctly or 'half correctly' and 13(15%) were misdiagnosed. Of the 95 recordings transmitted at 384kbit/s, 66 (69%) were diagnosed correctly or "half correctly" and 29 (31%) were misdiagnosed. |

ISDN, Integrated Services Digital Network; NR, not reported; USA, United States of America.

**Table 5.** Summary of studies using patient-operated teleultrasound devices.

| Study (year) | Sample size | Recruitment location | Communication and transmission | Summary of results |
|---|---|---|---|---|
| Axelrod (2025) [87] | 20 | Israel | • Internet<br>• Synchronous | Remote visits had a success rate of 97.4% (38 of 39), with significantly shorter durations compared with in-clinic visits (median 59.0 min vs. 159.0 min, $P < 0.001$). |
| Hadar (2022) [44] | 100 | Israel | • Internet<br>• Asynchronous | Remote ultrasound is a feasible solution for remote sonographic fetal assessment. Success in detection was 95.3% for fetal heart activity, 88.3% for body movements, 69.4% for tone, 92.2% for normal amniotic fluid volume, and 23.8% for breathing movements. Self-assessed user experience was rated at 4.4/5, whereas device satisfaction was rated at 3.9/5 using a Likert scale. |
| Le Vance [89] | 15 | United Kingdom | • Internet<br>• Asynchronous | In 24 remote, patient-operated scans, the fetal heartbeat, fetal movements, and an assessment of the liquor volume were identified in 92%, 83% and 100% of all ultrasound scans, respectively. 79% of all scans had all three criteria unanimously assessed. Acceptability was consistently rated high amongst participants. |
| Mor (2024) [65] | 50 | USA | • Internet<br>• Synchronous | Integrating routine home-ultrasound telemedicine visits into prenatal care can significantly reduce maternal anxiety during pregnancy and contribute to greater maternal attachment in individuals with a history of recurrent pregnancy loss ($p = 0.037$). |
| Nir (2024) [67] | 10 | Israel | • Internet<br>• Synchronous | Nine women (90%) were able to complete remote-modified biophysical profile assessment. Satisfactory amniotic fluid volume measurements were achieved in 100% of participants. The telemedicine encounter was significantly shorter (93.1 ± 33.1 min) than the in-person visit (247.2 ± 104.7 min; $p < 0.001$). |
| Pardo (2025) [92] | 107,167 | Israel | • Internet<br>• Asynchronous | A self-operated home ultrasound device, during the second and third trimester, is safe and not significantly associated with any pregnancy adverse event or neonatal complications ($p > 0.05$). The device was used in addition to routine antenatal care. |
| Pardo (2024) [93] | 28 | Israel | • Internet<br>• Synchronous and asynchronous | The fetal heart rate was measured using remote ultrasound in 84.7 ± 11.24% of scans made in app-guided mode and 96.3 ± 6.35% of scans made in clinician-guided mode. Corresponding values for the maximum vertical pool were 91.7 ± 2.31% and 95.0 ± 1.73%. The sensitivity for evaluating maximum pool depth was 87.5% and 100% in app-guided and clinician-guided modes. Specificity was 95% and 95.5% in both modes, respectively. |
| Pontones (2023) [70] | 46 | Germany | • Internet<br>• Asynchronous | Two-thirds of the women were able to imagine performing the self-guided examination at home, but 87.0% would prefer live support by a professional. Concerns about their own safety and that of the child were expressed by 23.9% of the women. Success rates for locating the target structure were 52.2% for videos of the fetal heartbeat, 52.2% for videos of the amniotic fluid in all four quadrants, and 17.9% for videos of the fetal profile. |

USA, United States of America.

However, the study sample size was small (35 fetuses), and postnatal diagnosis verification was limited. An earlier study by the same team, conducted on later gestation fetuses, achieved a complete cardiac examination within 86%–95% of cases (single observer with two independent operators) [80]. Contrastingly, a later study by Adriaanse and colleagues whereby three observers analyzed 10 second-trimester cases demonstrated a perfect agreement for pathology identification in only 20% of cases [25].

The specialist nature of these modalities represents their main weaknesses to widespread implementation and currently may be best suited as an adjunct rather than a replacement to two-dimensional ultrasound. A summary of studies can be seen in Table 6.

**Table 6. Summary of studies using three- and four-dimensional teleultrasound.**

| Study (year) | Sample size | Recruitment location | Communication and transmission | Summary of results |
|---|---|---|---|---|
| Adriaanse (2012) [25] | 10 | Netherlands | • NR<br>• Asynchronous | In a telemedicine setting using spatiotemporal image correlation (STIC) volumes, fetal cardiac anomalies can be diagnosed correctly by an expert. However, details required for adequate counseling and planning of postnatal care may be missed. STIC by telemedicine is a promising modality, although not accurate enough for exclusive use in clinical decision-making regarding treatment, prognosis, or termination of pregnancy |
| Arbeille (2014) [26] | 8 | Romania, French Guyana, Spain | • Internet<br>• Asynchronous | Cases scanned via 3-dimensional teleultrasound adequately visualized by the expert in seven of eight (88%) examinations of the fetal head, femur, and umbilical cord, and eight of eight (100%) examinations of the fetal abdomen and placenta |
| Ferrer-Roca (2006) [42] | 32 | Spain | • Internet and ISDN<br>• Asynchronous | Final volumes were small (1.5 megabits) and required about $4\pm2$ min to be transmitted over one channel (64 kilobits). In 30%, images were considered of low quality and in 29% of good quality; diagnosis could be done with confidence in all except seven cases. Limitations were linked to incomplete sampling due to the short acquisition time periods (26 s) and difficulties on hand-free probe movement. |
| Mabuchi (2020) [55] | 182 | Japan | • Internet<br>• Asynchronous | 21 datasets were excluded because of data unavailability or poor image quality. Congenital heart disease by 3-dimensional teleultrasound was detected in 14.9% of cases (24/161); the accuracy of prenatal diagnosis was 95.0% (153/161). |
| Michailidis (2001) [63] | 30 | United Kingdom | • Internet<br>• Asynchronous | Preliminary results demonstrate that a three-dimensional virtual examination of the fetal heart is possible. A complete heart examination was accomplished in 23 of 30 cases (76.7%). There are limitations such as the lack of flow and functional information, but complete ascertainment of the main cardiac connections was possible in the majority of cases. |
| Nelson (2001) [66] | 56 | USA | • Internet and courier<br>• Asynchronous | Overall, three-dimensional ultrasonography could produce diagnostic-quality results comparable with those of two-dimensional ultrasonography. Three-dimensional ultrasonographic image quality was lower than that of two-dimensional ultrasonography. Two- and three-dimensional ultrasonographic measurements were comparable (<5% difference) as was the extent of organ visualization, although some structures were challenging for both two- and three-dimensional ultrasonography. |
| Vinals (2008) [79] | 49 | Peru | • Internet<br>• Asynchronous | STIC volumes acquired between 11 + 0 and 13 + 6 weeks' gestation could be sent over the internet and their analysis enabled recognition of most of the structures and views necessary to assess the small fetal cardiac anatomy, with a high degree of interobserver concordance ($k>0.6$). |
| Vinals (2005) [80] | 50 | Chile | • Internet<br>• Asynchronous | STIC volumes can be obtained by operators inexperienced in fetal echocardiography, transmitted via the internet, and their analysis enables recognition of most of the structures and views necessary to assess fetal cardiac anatomy. The preliminary use of tele-STIC allowed us to demonstrate that some intracardiac anomalies can be ruled out and others confirmed, allowing perinatal management to be tailored accordingly. |

ISDN, Integrated Services Digital Network; NR, not reported; USA, United States of America.

**Robotic ultrasound.** The implementation of robotic obstetric teleultrasound remains theoretically plausible; however, the beneficial applications remain doubtful [23,24,26–28,91]. Studies demonstrated that transmission could be conducted across far distances (up to 7,000 km), primarily synchronously, with reasonable visualization of general anatomical structures [24, 27, 28]. Intraclass correlation between robotic teleultrasound and conventional ultrasound was high (>0.90) for the four main biometric measurements taken, reported by two studies. [23,91]. However, key anatomical features, such as the cavum septum pellucidum were only visualized in a third of cases using robotic ultrasound versus 100% of conventional two-dimensional ultrasound cases, as noted by Adams and colleagues [23]. Transmission delays were a common barrier for achieving adequate scans, resulting in lag on the expert's side, thus leading to poor visualization of anatomical structures [24, 28]. This attributed to an extended time for scan completion compared to conventional scanning [26,91]. Furthermore, operators of the robotic device itself were often inexperienced, leading to frequent repositioning of the scan probe and difficulties for the expert to achieve all the images required [27]. However, patient satisfaction was high, noting a reduction in long-distance travel as a main benefit of this technology [91]. A summary of studies using robotic teleultrasound can be seen in Table 7.

**Low- and middle-income countries.** In 10 studies, teleultrasound services were conducted in low- and middle-income countries across all gestations [6,40,41,50,52,79,81,84,90]. Low-cost tele-transmission services were feasible to develop with minimal overall loss to image quality [40,41]. Four studies examined point-of-care ultrasound (POCUS), utilized by healthcare professionals who were inexperienced with ultrasound, combined with tele-supervision by trained clinicians [50,52,81,84]. Studies frequently incorporated an in-depth multi-week training curriculum, enabling operators to develop a strong understanding of obstetric ultrasonography [50,52,81,84,85]. Operators were able to acquire a range of fetal anatomical assessments with high concordance when the same images were viewed by independent

**Table 7. Summary of studies using robotic teleultrasound.**

| Study (year) | Sample size | Recruitment location | Communication and transmission | Summary of results |
|---|---|---|---|---|
| Adams (2018) [23] | 30 | Canada | • Internet<br>• Asynchronous | Intraclass correlations showed excellent agreement (>0.90) between telerobotic and conventional measurements of all 4 biometric parameters. Sonographers generally reported that manipulating the mock transducer resulted in less physical strain than scanning a patient with a similar body habitus using a conventional ultrasound system. However, the patient site assistants reported that holding and grossly positioning the frame for the robotic arm caused moderate or severe physical strain in several cases. |
| Adams (2020) [24] | 18 | Canada | • Internet<br>• Synchronous and asynchronous | 11/21 cases were adequate for interpretation. Technical difficulties were experienced in 5/21 (24%) exams. Transmission delays of up to 10 s noted in 14% of scans. In each of these cases, there was a delay between the time the mock probe was repositioned and when the ultrasound interface displayed the new corresponding image. |
| Arbeille (2014) [26] | 8 | Romania, French Guyana. Spain | • Internet<br>• Asynchronous | Generating the two-dimensional images and selecting the sequence to be processed from the transmitted video took less than 3 min on average. Transmission delay and total processing ranged from four to 19 min. 93% of total studies were satisfactory for interpretation. |
| Arbeille (2005) [27] | 29 | Spain | • Telephone or satellite<br>• Synchronous | In 93.1%, all biometric parameters, placental location, and amniotic fluid volume, were correctly assessed. Transmission delay was 1 s and transmission failures occurred in approximately 9% of scans. Internet communication and robot activation setup took 10–15 min depending on the availability of the requested data flow and image frame rate through the satellite network. |
| Arbeille (2016) [28] | 15 | France | • Internet<br>• Synchronous | Image qualitative subjectively reported as similar to conventional ultrasound images. Transmission delay of up to three seconds when using ground internet. Image quality not affected using internet connections in different settings. |
| Olsen (2025) [91] | 46 | Norway | • Internet<br>• Synchronous | Biometric measurements showed excellent reliability (intraclass correlation coefficient between 0.990 and 0.993) with acceptable limits of agreement. Patient experiences were asked and 94% of the women scored for highest level of satisfaction. Interviews revealed the value of avoiding long-distance travel when image quality and digital communication experiences are good. |

obstetricians [50,84,85]. Continued tele-mentoring facilitated updated learning and continued clinical improvement and the adoption of interprofessional task sharing with capacity building enabled an improvement in ultrasound accessibility in pre-hospital sectors [52,81]. A consistent approach to continued verification of ultrasound scans and evaluation of outcomes by experienced staff facilitated a safe approach in areas with minimal technological infrastructure. The low cost of the devices was highlighted as a critical enabler for providing ultrasound services to economically strained countries [50,52]. Teleultrasound in low- and middle-income countries provides an avenue to improve access to maternity care in areas known to have disproportionately high adverse perinatal outcomes compared to higher-income countries. A summary of studies using teleultrasound in low- and middle-income countries can be seen in Table 8.

**Patient and provider experiences.** Five of the 61 aforementioned studies primarily evaluated patient and provider experiences with teleultrasound services [31,46,52,61,76]. Obstetric teleultrasound was associated with improved patient satisfaction, with many preferring subsequent teleultrasound scans, irrespective of the outcome of the antenatal scan findings [61]. Patients also cited high satisfaction for reduced travel to consultations [61,76,87,89]. Many studies examined patient satisfaction using a 5-point Likert scale, frequently reporting scores between 4 and 5 out of 5 [23,24,34,36,50,54,61,70,74,76,77,82,89].

Stakeholders cited benefits of teleultrasound in relation to timely identification of high-risk pregnancies and improved access to antenatal services, which enabled a balance in healthcare equity across obstetric communities [31,46,52]. Service users cited preference over synchronous versus asynchronous teleultrasound transmission [46]. Whilst initial challenges were noted by sonographers in relation to anxiety and pressure from real-time observation by senior clinicians, the iterative nature of the studies meant that barriers were addressed swiftly and reconfigured into beneficial factors [31]. Upskilling, increased access to timely specialist feedback and improved management of complex pregnancies were cited as the main benefits for service users [31,45]. Additional studies cited an increase in clinician confidence with teleultrasound using a Likert scale [25,34,46,50,60,84]. Clinician satisfaction was also rated highly among several studies [23,31,46,50,51,60,81]. A summary of qualitative studies can be seen in Table 9.

**Economic burden.** Four studies primarily evaluated economic outcomes, with additional studies examined cost outcomes alongside clinical outcomes [29,30,35,36,38,45,50,58,64,85,86]. Initial startup cost for adequate technological infrastructure was high, ranging from $10,355 to $101,750 [29,30,45,58,85]. The economic burden was noted to be higher for earlier published studies ($101,750 in 1998) [58], with more recent studies demonstrating a reduction in implementation costs of $10,355–$22,450 [29,85]. However, the number of scans performed in the earliest study was 600 per month across 3 centers [58], whilst more recently published studies included 804 in a single year [85], and 322 over 48 months [29]. Consequently, the frequency of scans performed is an important consideration when interpreting the economic burden. The cost of an individual teleultrasound encounter was frequently less than standard care; noted to be up to nine times less by Cuneo and colleagues [29,30,36,86]. Studies reported substantial overall monthly non-fixed cost savings using teleultrasound, enabling recovery of fixed implementation costs within 12–14 months [35,58]. Dowie and colleagues demonstrated that the cost of a telemedicine encounter was greater than face-to-face consultation (£206 versus £74), however, within 14 days overall costs were neutral due to a reduction in travel costs [38]. The clinical application of the included studies, such as antenatal screening or antenatal diagnosis, did not seem to substantially influence the economic burden of teleultrasound implantation. Teleultrasound may be a viable option for antenatal care, however, high initial costs need to be mediated by offering the service to a wide cohort of women and for an extended duration. A summary of economic studies can be seen in Table 10.

A summary of the overall study findings, GRADE assessments, and future recommendations are in Table 11.

## Discussion

This systematic review and meta-analysis was designed to comprehensively assess the role of teleultrasound for obstetric care by evaluating feasibility, diagnostic accuracy, acceptability, utility, and economic burden. Despite included studies

**Table 8. Summary of studies recruiting in low-and middle-income countries.**

| Study (year) | Sample size | Recruitment location | Communication and transmission | Summary of results |
|---|---|---|---|---|
| Ahmed (2024) [86] | 513 | Egypt and one other country (not specified) | • internet<br>• Synchronous | Tele-echocardiography and counseling were successful in all the cases. Satisfaction with the service was 3.8/4, with the main limitation being the need for further referral to a tertiary center for delivery. |
| Ferlin (2012) [40] | 20 | Brazil | • internet<br>• Synchronous and asynchronous | Transmission delays were one to three seconds. While all fetal structures could be visualized and identified, the observers believed that some aspects in relation to the size of the field of view, focal zone, gray scale or text information were impaired. The use of low-cost software appeared suitable for screening for chromosomal abnormalities in the first trimester of pregnancy. |
| Ferreira (2014) [41] | 15 | Brazil | • internet<br>• Synchronous | Tele-ultrasound transmission of fetal central nervous system structures using an inexpensive video streaming device provided images of subjective good quality. The generation and interpretation of ultrasound images are still very user-dependent and require a highly skilled operator to obtain and interpret the images. |
| Jemal (2024) [50] | NR (100 scans) | Ethiopia | • internet<br>• Synchronous | Healthcare provider–performed antenatal ultrasound, supported by obstetricians via tele-ultrasound within Ethiopia showed high levels of concordance, was well received by participants and provided the area with enhanced access to antenatal imaging. |
| Kumar (2023) [52] | 61 | Kenya | • internet<br>• Asynchronous | The use of POCUS with telemedicine to enable sonographer review of images and data captured by nurses can enable effective regulation of the service—a key step towards scaling and sustainability. |
| Kozuki (2016) [85] | 804 | Nepal | • internet<br>• Asynchronous | Following a multi-week training program, primary-level healthcare workers in rural Nepal can accurately diagnose selected third-trimester obstetric risk factors using ultrasonography when the images were subsequently assessed by radiologists. |
| Nieto-Calvache [90] | 60 | Argentina, Brazil, Colombia, Egypt, England, Ghana, Indonesia, Ireland, Italy, Taiwan, USA | • internet<br>• Asynchronous | Teleconsultant antenatal evaluation of placenta accreta spectrum, and management plans matched those of the local team in 71.7% of the cases. The severity of placenta accreta was overestimated in nine reviews (16.9%) and was underestimated in six reviews (11.3%). |
| Toscano (2021) [84] | 126 | Peru | • internet<br>• Asynchronous | This Obstetric ultrasound tele-diagnostic system is a promising means to increase access to diagnostic Obstetric ultrasound in low-resource settings. The tele-diagnostic system demonstrated excellent agreement with standard of care ultrasound. Fetal biometric measurements were acceptable for use in the detection of gross discrepancies in fetal size requiring further follow-up. |
| Vinals (2008) [79] | 49 | Peru | • internet<br>• Asynchronous | Using a telemedicine link via the internet is technically feasible. The spatiotemporal image correlation (STIC) volumes acquired between 11 + 0 and 13 + 6 weeks' gestation could be sent over the internet and their analysis enabled recognition of most of the structures and views necessary to assess the small fetal cardiac anatomy, with a high degree of interobserver concordance. The success of telemedicine depends mainly on the quality of the acquired volume datasets |
| Vinayak (2018) [81] | NR (271 scans) | Kenya | • internet<br>• Asynchronous | Collaborative interprofessional task sharing between radiologists, sonographers, and midwives using POCUS and tele-transmission is a viable solution to current healthcare resource shortages in sub-Saharan Africa and other LMIC settings. The interprofessional task sharing initiative provided a platform for the establishment of a collaborative model of care that effectively addressed geographical issues of health service access along with the provision links and pathways for referring for detected high-risk scenarios |

LMICs, low-and middle-income countries; NR, not reported; POCUS, point-of-care ultrasound; USA, United States of America.

**Table 9. Summary of included qualitative studies.**

| Study (year) | Sample size | Recruitment location | Communication and transmission | Summary of results |
|---|---|---|---|---|
| Bidmead (2020) [31] | 23 | United Kingdom | • Internet<br>• Synchronous | Service users and clinical stakeholders were accepting of the teleultrasound service. Service users reported satisfaction with communications during the consultation and awareness that telemedicine had facilitated local access to clinical expertise. Whilst clinical stakeholders reported challenges, the iterative nature of the evaluation meant that concerns were discussed, responded to, and overcome as the pilot developed. Clinical stakeholders' perception of benefits for service users encouraged their acceptance. |
| Hishitani (2014) [46] | 14 | Japan | • Internet<br>• Synchronous and asynchronous | The introduction of fetal telediagnosis significantly increased staff confidence in performing fetal cardiac screening (score 2.3 at start, 3.4 at study completion; $p = 0.034$). The rate of score increase rose with the number of telediagnoses ($p < 0.05$). Real-time image transmission was preferred over recorded images (score 3.7 vs. 2.4, respectively; $p = 0.042$). |
| Kumar (2023) [52] | 61 | Kenya | • Internet<br>• Asynchronous | Nurse-led obstetric POCUS with tele-transmission improved access and affordability of obstetric ultrasonography services, timely identification and referral of high-risk pregnancies, and improving awareness of appropriate antenatal care among underserved populations. The relative affordability of the POCUS device was described as a critical enabler for a business model targeting low- and middle-income segments of the population, and for increasing quality and equity of antenatal care coverage. |
| McCrossan (2012) [61] | 66 | Northern Ireland | • ISDN<br>• Synchronous | Participants expressed very high satisfaction rates with fetal telecardiology, equivalent to face-to-face consultation. The telecardiology appointments were associated with significantly reduced travel times and days off work ($p < 0.01$). Expectant mothers expressed a clear inclination for a fetal cardiology appointment at the local hospital facilitated by telemedicine ($p < 0.01$). |
| Smith (2021) [76] | 297 | United Kingdom | • Internet<br>• Synchronous | Overall, women expressed high levels of satisfaction with the telemedicine consultation. Travel to the telemedicine consultation took a median time of 20 min, in comparison to an estimated journey of 230 min to the fetal medicine center. |

ISDN, Integrated Services Digital Network.

**Table 10. Summary of included economic evaluation studies.**

| Study (year) | Sample size | Recruitment location | Communication and transmission | Summary of results |
|---|---|---|---|---|
| Beldjerd (2023) [30] | 260 | France | • Internet<br>• Asynchronous | The expected average total cost for tele-expertise for a patient was €74.45 (95% CI: €66.36–€82.54) compared to €195.02 (95% CI: €183.90–€206.14) for the conventional face-to-face strategy. Accordingly, using tele-expertise led to a statistically significant reduction of €120.57 in the average total cost per patient ($p < 0.05$). A sensitivity analysis confirmed the robustness of the model produced. |
| Dowie (2008) [38] | 76 | United Kingdom | • Internet<br>• Asynchronous | A telemedicine assessment of 5 min duration was more costly than an examination in London (mean cost per referral of £206 vs. £74, $p < 0.001$). However, the telecardiology service was cost-neutral after 14 days and for the extended period until delivery. Travel costs for London women averaged £37 compared with £5.50 for the telemedicine referrals. |
| Malone (1998) [58] | NR | USA | • Internet<br>• Synchronous and asynchronous | Monthly non-fixed cost savings by eliminating videotape review include $1,620 to $2,700 for printing still images, $1,200 for courier charges, and $7,000 for fewer repeat ultrasound examinations. Monthly non-fixed costs for the telemedicine network are $2,415. Net monthly savings in non-fixed costs for a telemedicine network are therefore $7,405 to $8,585, which may pay for the initial fixed costs in 12–14 months. |
| Mistry (2013) [64] | NR | United Kingdom | • Internet<br>• Synchronous | The probabilistic results from the model show that offering telemedicine to all standard-risk women is the dominant strategy, i.e., that the costs are lower and the QALYs are higher (that is, telemedicine is more effective). The sensitivity analyses found that the model was robust, and that telemedicine remained the most cost-effective strategy. |

USA, United States of America.

**Table 11. Summary of existing evidence.**

| Study type | | | |
|---|---|---|---|
| **Feasibility** | **Diagnostic accuracy** | **Qualitative** | **Economic evaluation** |
| **Evidence**: Feasible to transmit teleultrasound data synchronously and asynchronously across a wide range of distances. Transmission failures were minimal but infrequently commented on. Transmission delays were substantially longer in asynchronous studies. Overall, image quality was similar to the reference standard. However, a wide variation in successful identification of anatomical features for complex tele-scans was seen. Lower bandwidths demonstrated worse image quality. However, this may be suitable for basic fetal tele-scans. Synchronous teleultrasound may require higher bandwidth requirements so as not to compromise on image quality. | **Evidence**: Diagnostic accuracy was variable in included studies, but many cited >90%. Area under the curve was 0.93 with moderate sensitivity and moderate heterogeneity noted. Better image quality was not associated with improved accuracy. Inter-agreement was frequently moderate to excellent. Low agreement levels related to more subjective ultrasound features. Limited information on intraclass correlation noted. Sensitivity and specificity were generally high in included studies, demonstrating comparability to conventional ultrasound. | **Evidence**: Teleultrasound was associated with improved patient satisfaction. Corresponding with the reduced requirement for travel. Service users cited an increased confidence with teleultrasound and high satisfaction levels. Benefits of teleultrasound related to improved access to obstetric services, a balance in healthcare equity and improved care for underserved communities. | **Evidence**: Initial startup costs were high but frequently reclaimed back over the months following implementation due to monthly non-fixed cost savings. Teleultrasound appointments frequently cost less than standard care appointments. Initial evidence to suggest an improvement in future quality-adjusted life years for patients using teleultrasound services. |
| **GRADE assessment:** ⊕⊕⊕◯ Low[a] | **GRADE assessment:** ⊕⊕◯◯ Low[b] | **GRADE assessment:** Not applicable | **GRADE assessment:** Not applicable |
| **Recommendation:** Further research required to assess the minimum bandwidth and framerate requirements to successfully transmit high-quality synchronous teleultrasound data. | **Recommendation:** Large RCTs are required to compare diagnostic accuracy for prenatal diagnosis of teleultrasound vs. conventional ultrasound. Reference standard should be postnatal diagnosis. | **Recommendations:** Minimal recommendations required. | **Recommendation:** Full health economic analysis required, likely in conjunction with RCT. |

**Teleultrasound variant**

| Patient-operated | 3-Dimensional | 4-Dimensional | Robotic | Transvaginal | Simulation |
|---|---|---|---|---|---|
| **Evidence**: Small, but increasing number of studies included. Potential to perform a basic assessment of fetal well-being with high patient acceptability. Evidence to reduce patient anxiety noted. Limited scope at present to replace current standard of obstetric care. Limited assessment of economic burden explored. | **Evidence**: Image quality poorer compared to conventional ultrasound. Longer duration required for image pre-processing, resulting in transmission delays. Limited assessment of diagnostic accuracy, however, when noted interclass correlation was moderate. | **Evidence**: Feasibility demonstrated in asynchronous transmission. Variation in observer agreement noted within included studies. Specialist interpretation required representing its main weakness for wider implementation. | **Evidence**: Potential to transmit synchronously, but difficulties with transmission delays, lag, and poor image quality noted due to high bandwidth requirements. Further reliance on operators being experienced to handle probe. Beneficial application remains doubtful. | **Evidence**: Troublesome to separate transvaginal and transabdominal ultrasound data. Image degradation reported by a single study was minimal. | **Evidence**: No evidence found within this review. |

*(Continued)*

**Table 11.** (Continued)

## Study type

| Feasibility | Diagnostic accuracy | Qualitative | Economic evaluation |
|---|---|---|---|
| **GRADE assessment:** ⊕⊕○○ Low[c] | **GRADE assessment:** ⊕○○○ Very Low[d] | **GRADE assessment:** ⊕○○○ Very Low[f] | **GRADE assessment:** Not assessable. |
| **Recommendation:** Advancement in technology warranted. Incorporation of Doppler assessment would be of high impact. If possible, further feasibility studies are required with potential to conduct an RCT using such a device as a supplement to aspects of antenatal care. | **Recommendation:** Further theoretical and feasibility work required to determine superiority. May play a role in off-line assessment in remote locations whereby specialist access is scarce. | **GRADE assessment:** ⊕○○○ Very Low[e] **Recommendation:** Additional feasibility studies, comparing to other transmission modes required. Currently only suitable as an adjunct to conventional ultrasound. | **GRADE assessment:** Not assessable. |
| | | **Recommendation:** Further feasibility studies required. | **Recommendation:** Feasibility studies required. |

## Setting

| Home | Community | Low- and middle-income countries |
|---|---|---|
| **Evidence:** Six studies, including one randomized controlled trial assessed the application of teleultrasound in the home setting. Patient-operated devices demonstrated high ratings for acceptability and evidence of reduced anxiety. Minimal assessment of diagnostic accuracy outcomes with comparator groups, including technological requirements for successful transmission. Very limited assessment of patient-operated ultrasound in ethnic minorities and low education level. | **Evidence:** Limited number of studies. Feasibility of teleultrasound demonstrated in community/pre-hospital sectors. Assessment of diagnostic accuracy was high in community/pre-hospital sectors. Assessment of diagnostic accuracy was high when combined with tele-mentoring. Interprofessional task sharing improved teleultrasound accessibility within community/pre-hospital sectors. Low-cost devices are critical enablers to providing ultrasound services to economically strained countries. | **Evidence:** Low-cost teleultrasound services were feasible to develop with minimal loss to image quality. POCUS can be used by minimally trained operators, combined with tele-supervision. Limited assessment on diagnostic accuracy outcomes, however, when reported, concordance between independent observers was high. Low-cost devices are a critical enabler to providing ultrasound services to economically strained countries. |
| **GRADE assessment:** ⊕⊕○○ Low[g] | **GRADE assessment:** ⊕○○○ Very Low[h] | **GRADE assessment:** ⊕○○○ Very Low[i] |
| **Recommendation:** Further feasibility and diagnostic accuracy studies warranted until further conclusions can be made. | **Recommendation:** Further diagnostic accuracy studies required with emphasis on the technological infrastructure required. Potential for definitive RCT in the future. | **Recommendation:** Further feasibility studies required for implementation of more advanced teleultrasound infrastructure. Further pilot studies for POCUS with future definitive RCT. |

**GRADE framework for rating the strength of evidence**

- **High certainty:** available evidence provides a high level of confidence, and it is unlikely that further research will significantly change confidence in the estimate.
- **Moderate certainty:** available evidence is sufficient to support a conclusion, but further research may still impact confidence in the estimate.
- **Low certainty:** available evidence is limited, and additional research is likely to have an important impact on confidence in the estimate.
- **Very low certainty:** available evidence is insufficient to support any firm conclusions, and future research is expected to have a significant impact on confidence in the estimate.

*(Continued)*

**Table 11.** (Continued)

Explanations

- [a]The proportion of information from studies at high risk of bias is sufficient to affect the interpretation of results. All data acquired from observational studies, with slight inconsistency noted in the reporting of results. Observational studies frequently lacking a control group, leading to difficulty assessing the effects of an intervention. Variation in outcomes assessed, tele-transmission details, type of ultrasound performed, and participant eligibility criteria seen. Unable to clearly assess imprecision due to narrative description of results.

- [b]The proportion of information from studies at high risk of bias is sufficient to affect the interpretation of results. Moderate inconsistency was noted due to variable reporting of results. Frequently data was provided by observational studies which are of lower quality compared to randomized controlled trials. Unable to clearly assess imprecision due to narrative description of results. Variation in participant eligibility, tele-ultrasound frequency and control group criteria is seen.

- [c]The proportion of information from studies at high risk of bias is sufficient to affect the interpretation of results. Data was acquired from one randomized controlled trial. Observational studies frequently lacking a control group, leading to difficulty assessing the effects of an intervention. Variation in outcome reporting, ultrasound frequency and eligibility criteria seen. Unable to clearly assess imprecision due to narrative description of results.

- [d]The proportion of information from studies at high risk of bias is sufficient to affect the interpretation of results. Small number of included studies with control groups therefore difficult to clearly determine the observed effects due to the intervention. Substantial variation in the reference standards seen and the eligibility of included participants. Unable to clearly assess imprecision due to narrative description of results.

- [e]The proportion of information from studies at high risk of bias is sufficient to affect the interpretation of results. Moderate variation in tele-transmission type and participant eligibility is seen. Minimal observations studies with an available control group, therefore introducing difficulty to assess the effects of the intervention. Unable to clearly assess imprecision due to narrative description of results.

- [f]The proportion of information from studies at high risk of bias is sufficient to affect the interpretation of results. Moderate variation in tele-transmission type and participant eligibility is seen. Minimal observations studies with an available control group, therefore introducing difficulty to assess the effects of the intervention. Unable to clearly assess imprecision due to narrative description of results.

- [g]Minimal number of included studies. Two included studies, with one randomized controlled trial rated low risk of bias. Unable to clearly assess imprecision due to lack of and narrative description of results.

- [h]The proportion of information from studies at high risk of bias is sufficient to affect the interpretation of results. Moderate variation in tele-transmission type and participant eligibility is seen. Minimal observations studies with an available control group, therefore introducing difficulty to assess the effects of the intervention. Unable to clearly assess imprecision due to narrative description of results.

- [i]The proportion of information from studies at high risk of bias is sufficient to affect the interpretation of results. Data acquired from observational studies only. Small number of included studies with control groups therefore difficult to clearly determine the observed effects due to the intervention. Substantial variation in the reference standards seen, tele-transmission type, eligibility of included participants and monitoring frequency. Unable to clearly assess imprecision due to narrative description of results.

POCUS, point-of-care ultrasound; RCT, randomized controlled trial.

demonstrating a high risk of bias, primarily due to methodological heterogeneity, obstetric teleultrasound was feasible in a wide range of settings. This included both synchronous and asynchronous transmission. However, at times the image quality was insufficient to clearly assess finer anatomical structures in more complex fetal anatomical tele-scans. This was in conjunction with bandwidth requirement, which was a crucial factor to enable adequate image quality and reduced transmission delay. Higher bandwidth requirements were more commonly seen in newer published studies, and feasibility studies including a clinical focus on improving access to specialist expertise in remote and deprived areas. However, overall meta-analysis between obstetric teleultrasound and the reference standards was non-inferior for identifying fetal and placental structures. The diagnostic accuracy of teleultrasound was highly discriminative (AUC 0.93). The false positive rate was 0.03 demonstrates teleultrasound is an excellent tool for low-risk cases. The sensitivity was moderate (0.4 (95% CI [0.44,0.84]), with wide CIs, suggesting teleultrasound may miss potential high-risk diagnoses if used as a sole screening tool. Initial results suggest novice ultrasound users can competently perform obstetric scans using low-cost devices with concurrent tele-supervision. Patient-operated teleultrasound devices were emerging in the literature, with initial assessments of basic fetal well-being performed in the home setting. Examination of ultrasound variants (robotic, three-dimensional, and four-dimensional teleultrasound) did not highlight a superiority to two-dimensional ultrasound. The specialist nature of these tele-scan types represents the main limitation for widespread implementation.

Overall acceptability ratings for obstetric teleultrasound were high for both patients and providers, citing common benefits in relation to satisfaction, reduced travel, economic savings, increased access to obstetric care and balance in healthcare equity. Whilst start-up costs for teleultrasound are high, overall costs were less in more recently published studies and could normally be accrued back over the subsequent months due to overall capital savings from the teleultrasound service versus routine care.

There was an underrepresentation of high-quality randomized controlled trials, suggesting a need for further research. Furthermore, current literature is limited in the reporting of clear methodological and technological capabilities of the teleultrasound systems, proving difficulty for researchers to replicate studies. Despite this, obstetric teleultrasound for antenatal care has flourished over recent years, demonstrating a growing body of evidence to support the digital migration of pregnancy care. Additionally, the evolving use of obstetric teleultrasound in community and home settings further supports one of the three main pillars of the recently published 10-year plan provided by the national health service in the UK [94]. Consequently, to support national objectives and future implementation, it is imperative for studies, particularly randomized controlled trials to consider a more standardized approach to study design, particularly regarding type of antenatal ultrasound, gestational age at time of ultrasound, the reference standard and clinical outcomes assessed. This will develop a greater cohesive and applicable body of evidence for obstetric teleultrasound which would enhance quantitative analysis and potentially aid in wider implementation.

Findings in this review align closely with previously published narrative literature on the use of teleultrasound for obstetric care [3,95,96]. The most recent review by Kariman and colleagues explores the use of obstetric ultrasound within 31 studies, highlighting similar findings to our review in relation to feasibility and acceptability of teleultrasound, however, included a more focused approach on point-of-care devices without incorporated teleultrasound [3]. Similarly to our review, a lack of high-level evidence was available, further highlighting a clear research requirement for future research, particularly assessing its impact on clinical maternal-fetal outcomes. Whilst the evaluation of teleultrasound on clinical outcomes such as adverse perinatal outcomes would be clinically useful, this may be troublesome given its rare occurrence, requiring a very large sample size to suitably assess. Therefore, usage of a composite perinatal outcome score, may be more preferable in future prospective studies; a factor that has been previously incorporated in recent high-level randomized controlled trials exploring alternative telemedicine obstetric interventions [97]. Vitally, alongside investigating feasibility and diagnostic accuracy, researchers should assess the impact of teleultrasound on other key service-led outcomes such as emergency hospital attendance and hospital admission, enabling a thorough understanding of the safety of such interventions. A health economics analysis is also encouraged to formally assess the economic burden of teleultrasound interventions.

In conjunction, teleultrasound usage has the potential to raise several ethical and regulatory issues which are imperative to consider when globally assessing obstetric teleultrasound functionality future high-evidenced studies, and the potential for wider rollout. Firstly, a detailed informed consent process is imperative, ensuring patients attain clarity on who will be interpreting the ultrasound scan if it is digitally transmitted. This is particularly important in asynchronous systems whereby the interpreting clinician may not be physically present and subsequent immediate explanation of the scan results is somewhat limited. This may lead to reduced patient satisfaction if this is not clearly explained during the consent process. Secondly, the reporting of essential technology security standards such as data encryption, data storage/retention, and liability in cross-jurisdictional teleultrasound usage is vital. Transmission of data should be watertight to minimize any breach of sensitive information. Data stored on cloud or shard databases must comply with data protection regulations such as the Health Insurance Portability and Accountability Act (HIPAA) in the USA and General Data Protection Regulation (GDPR) in Europe. Whilst cross-jurisdictional teleultrasound transmission may be necessary for certain study/clinical settings, different privacy laws may apply, complicating compliance and liability, particularly in uncommon situations where a fetal abnormality is missed. Thirdly, cost and reimbursement is an essential issue to consider when using teleultrasound on a wider scale. Insurance and national health systems may not yet have clear frameworks for teleultrasound reimbursement. This can limit equitable access to teleultrasound services and should be address prior to wider implementation. Maintaining professional standards and training in teleultrasound is also key to maximizing diagnostic and technical capabilities. The competency of remote operators requires continued training, whilst professional bodies (e.g., RCOG) should develop standards for teleultrasound practice, such as minimum technical requirements and quality assurance measures, This is a key facet for wider scale teleultrasound implementation. Finally, continued auditing of teleultrasound services is required, particularly in relation to clinical outcomes, patient satisfaction, and data portion. This is essential to maintaining ethical integrity.

Key themes in prior reviews suggest the need to produce a low-cost affordable teleultrasound system which enables clinicians to reduce the geographical and ethnic disparities that can be seen in obstetric care [3,89,90,95,96]. This is particularly important in low- and middle-income countries where the disparities in poorer obstetric outcomes are clear. There is some evidence to suggest that teleultrasound may benefit low- and middle-income countries, however, prior generalized reviews substantially underrepresent obstetric studies, suggesting further research in this area was required [8]. Our updated review provides insight into the recent work currently undertaken, particularly incorporating long-distance synchronous tele-transmission in pre-hospital settings using tele-supervision. In the absence of capacity to implement advanced healthcare infrastructure in such areas, the addition of basic telesonography services may prove beneficial and improve patient access to maternity care [4]. However, simple infrastructure requirements such as continuous electricity and costs for tele-system maintenance need to be considered and are perceived barriers to remote geographical implementation [98]. Importantly, almost half of the world's population reside in remote areas, which can be susceptible to substandard telecommunication coverage [99]. Recent data suggests that even the least developed countries are capable of internet bandwidth connections of between 256 Kb/s and 2 Mb/s, suggesting teleultrasound implementation on the most basic level is plausible [100]. Needless to say, obstetric ultrasound training and subsequent regulation of antenatal teleultrasound services should be rigorous to reduce risk of harm secondary to improper use in this context. Therefore, measures to ensure consistent quality control should be integrated into future study design within low- and middle-income countries [4,101].

Interestingly, the integration of artificial intelligence (AI) into obstetric ultrasound has been explored. Blind abdominal sweeps using a low-cost device has recently been assessed in Zambia for first-trimester scanning, whereby accuracy was deemed similar to scans performed by trained sonographers [102]. This alludes to an alternative avenue for providing obstetric care in more deprived countries. However, clinicians incorporating AI must acknowledge certain technology-specific regulatory factors prior to wider implementation. This includes, but is not limited to, responsibility in the presence of an error, the requirement for continued technology validation, ensuring consistency of data production in the presence of diverse populations and ownership of the data that is produced.

This review contained a balance of synchronous and asynchronous transmission services. Asynchronous obstetric teleultrasound transmission may be more suitable on a global scale, as images can be transmitted over lower bandwidths and less reliable telecommunication channels, as concerns regarding transmission delays/lag become less important [4]. Asynchronous transmission may be suitable for scans requiring expert input, which can be infrequently attained within a timely fashion [103]. However, in the context of obstetrics, an emergency intervention may be warranted at times following antenatal ultrasound scanning. Therefore, synchronous transmission may be preferable for the majority of obstetric cases. Additionally, synchronous transmission facilitates a timely interaction between the patient and operator which fosters collaborative discussion surrounding subsequent antenatal care management and supports a patient-centered care [34,52]. In areas whereby telecommunication services are unequipped to transmit full quality images via synchronous transmission, an alternative approach is Remote Tast Scale [4]. This encompasses synchronous transmission of low-quality images followed by asynchronous transmission of high-quality images. No studies in this review explored this option, however, prior literature has examined this transmission type [5].

Crucially, the implementation of telehealth services is frequently in relation to clinician attitudes for endorsement. Qualitative assessment is limited in prior reviews, but initial insight in this review suggests high provider acceptability, suggesting a potential turning point in obstetric teleultrasound adoption [95,96]. Endorsement from both patients and clinicians regarding the novel emergence of remote patient-operated teleultrasound devices is high. Patient-operated devices may be a potential avenue for digital innovation which could modify how current antenatal care pathways are delivered [104,105]. However, presently, the device capabilities are limited to only simple assessments of fetal wellbeing [44,65,67,70,87,89,92,93]. The adjunct of AI and integration of sophisticated fetal assessments, such as remote biometry and Doppler, may enable clinicians to sophisticaly monitor patients at home and reduce the capacity concerns within current outpatient services.

Irrespective of technological advancements: a lack of digital literacy, presence of language barriers, and distrust in healthcare services by patients will remain as the main inhibitors to implementation. Such factors are more prominent in lower socioeconomic groups and ethnic minorities [106–108]. Therefore, it is imperative for future studies to support these patient groups, ensuring to promote patient education and empowerment via suitable teaching strategies and collaboration with interpreter services. Synchronous transmission may be a preferable option for such patient groups, particularly in the case of patient-operated teleultrasound devices.

Fig 9 represents a selection of barriers and/or facilitators for teleultrasound implementation, which have been stratified based on the evidence consolidated from this review. Minimization of barriers and adoption of facilitators may enable clinicians to achieve important beneficial teleultrasound outcomes listed at the bottom of the figure.

The extensive search, including gray literature and subsequent evaluation of a wide range of factors regarding teleultrasound represent the main strengths of this review. Importantly, this review enables clinicians and researchers to understand the current state of teleultrasound and determine how best this technology can be safely implemented into obstetric care. This review further facilitates clinicians to consider the technological requirements for teleultrasound optimization, whilst determining the elements required for robust future research. Furthermore, included studies derived from a range of countries, thus improving the generalizability of results.

Due to the variable quality of data and methodological heterogeneity of the included studies, meta-analysis was limited, particularly pairwise meta-analysis. A lack of consistent outcome reporting currently limits a deep quantitative analysis of obstetric teleultrasound. Additionally, due to the experimental nature of teleultrasound, it is plausible that many studies yielding negative results were never formally reported. From a technological perspective, this may be less important, given that the included studies within this review were sufficient to evaluate the beneficial and negative technological applications of teleultrasound. However, the presence of publication bias may have a greater impact on more serious outcomes, such as patient safety and adverse perinatal outcomes, which were difficult to assess due to the few studies included using obstetric teleultrasound within a clinical context. It is important to rationalize these

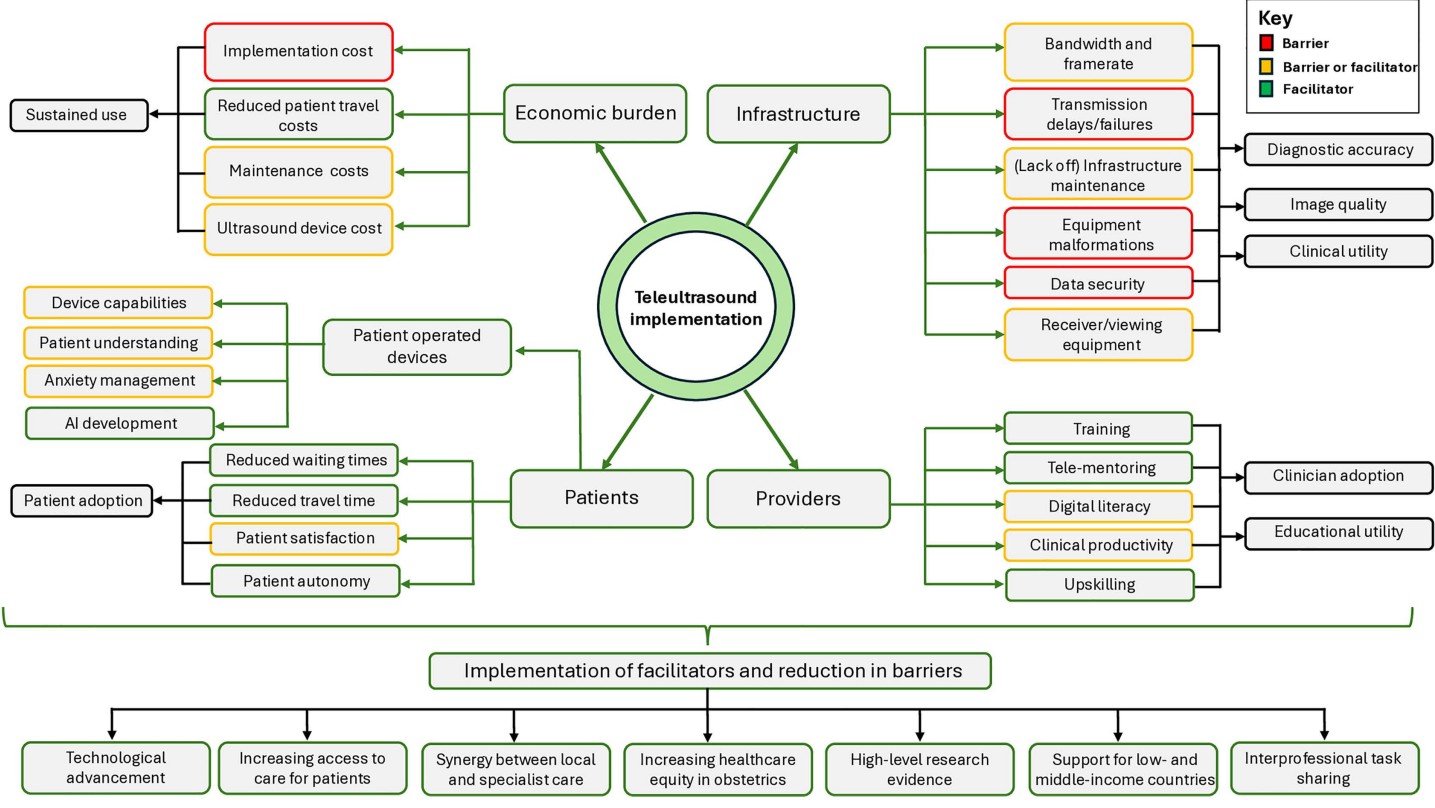

**Fig 9. A schematic diagram illustrating several barriers and facilitators to achieving successful teleultrasound implementation.** AI, artificial intelligence.

areas for future use and further studies are warranted to include all domains for adequate quality-of-care evaluation [109]. Finally, 21% of included studies were more than 20 years old. Given the ever-evolving digital landscape and escalation in technology sophistication, the age of these studies may limit the relevance of their results to modern obstetric practice.

This review has demonstrated the potential applicability and value of teleultrasound for obstetrics across a range of outcomes. The rapid development of teleultrasound services aimed to address the current capacity concerns within outpatient obstetric services and the subsequent impacts this has on patients and providers. This novel care model is everchanging and new ultrasound devices capable of telesonography are of clinical and scientific relevance. It is imperative that studies ensure sufficient methodological detail and consistent outcome reporting to inform future study design. Presently, additional high-quality evidence is required, particularly using obstetric teleultrasound within a clinical context before recommendations can be made regarding teleultrasound as an alternative avenue for antenatal care.

## Supporting information

**S1 Table. PRISMA checklist.** From: Page MJ, McKenzie JE, Bossuyt PM, Boutron I, Hoffmann TC, Mulrow CD, and colleagues. The PRISMA 2020 statement: an updated guideline for reporting systematic reviews. BMJ 2021;372:n71. https://doi.org/10.1136/bmj.n71. This work is licensed under CC BY 4.0. To view a copy of this license,

**S2 Table. Search strategy.**
(DOCX)

**S3 Table. Critical appraisal checklist for observational studies.**
(DOCX)

**S4 Table. Excluded studies and associated reasons for exclusion.**
(DOCX)

**S5 Table. Critical appraisal results for all studies.**
(DOCX)

**S6 Table. Description of study methodology.**
(DOCX)

**S7 Table. Diagnostic accuracy of teleultrasound within reporting studies, stratified by indication.**
(DOCX)

**S1 Fig. Single variable proportional meta-analysis of identification rates for fetal and placental structures using teleultrasound, stratified by body system.**
(PDF)

**S2 Fig. Single variable proportional meta-analysis of diagnostic accuracy rates for teleultrasound.** (*) represents the reference standard was conventional in-hospital ultrasound. (^) represented the reference standard was postnatal diagnosis.
(TIF)

## Author contributions

**Conceptualization:** Jack Le Vance.

**Data curation:** Jack Le Vance, Matthew Vaughan, Tanvi Bhatia, Leo Gurney.

**Formal analysis:** Jack Le Vance, Matthew Vaughan, Tanvi Bhatia, Leo Gurney.

**Investigation:** Jack Le Vance, Matthew Vaughan, Tanvi Bhatia.

**Methodology:** Jack Le Vance, Matthew Vaughan, Tanvi Bhatia, Leo Gurney.

**Supervision:** Leo Gurney, Victoria Hodgetts Morton, R.Katie Morris.

**Validation:** Leo Gurney, Victoria Hodgetts Morton, R.Katie Morris.

**Writing – original draft:** Jack Le Vance.

**Writing – review & editing:** Jack Le Vance, Matthew Vaughan, Tanvi Bhatia, Leo Gurney, Victoria Hodgetts Morton, R.Katie Morris.

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
