## [Editor Report · Decision Letter 0]

11 Sep 2025

Dear Dr Le Vance,

Thank you for submitting your manuscript entitled "Teleultrasound in obstetrics: a systematic review and meta-analysis" for consideration by PLOS Medicine. Apologies for the delay in providing you with an initial decision.

Your resubmission has now been evaluated by the PLOS Medicine editorial staff as well as by an academic editor with relevant expertise and I am writing to let you know that we would like to send your manuscript out for external peer review.

For clinical studies, please upload a copy of your trial study protocol as a supporting information file. The study protocol should be the version submitted for approval to the institutional review board or ethics committee, should include any amendments to the study protocol, as well as the date of their approval by the institutional review or ethics committee. Please also detail any deviations from the study protocol in the Methods section of your manuscript. The editors will consider the protocol and study conduct prior to a final decision for external review.

Please re-submit your manuscript within two working days, i.e. by Sep 15 2025.

Feel free to email me at atosun@plos.org or us at plosmedicine@plos.org if you have any queries relating to your submission.

Kind regards,

Alexandra Tosun, PhD

Senior Editor

PLOS Medicine

---

## [Decision Letter · Decision Letter 1]

3 Nov 2025

Dear Dr Le Vance,

Many thanks for submitting your manuscript "Teleultrasound in obstetrics: a systematic review and meta-analysis" (PMEDICINE-D-25-03073R1) to PLOS Medicine. The paper has been reviewed by subject experts and a statistician; their comments are included below and can also be accessed here: [LINK]

Please note that we have invited some of the same reviewers who evaluated the previous version of the manuscript and invited new reviewers as well.

As you will see, the reviewers find that the meta-analysis strengthened the paper. However, they still have some additional questions and concerns, e.g. about the inclusion and exclusion of studies. After discussing the paper with the editorial team and an academic editor with relevant expertise, I'm pleased to invite you to revise the paper in response to the reviewers' comments. We plan to send the revised paper to some or all of the original reviewers, and we cannot provide any guarantees at this stage regarding publication.

We ask that you submit your revision by Nov 24 2025. However, if this deadline is not feasible, please contact me by email, and we can discuss a suitable alternative.

Don't hesitate to contact me directly with any questions (atosun@plos.org).

Best regards,

Alexandra

Alexandra Tosun, PhD

Senior Editor

PLOS Medicine

atosun@plos.org

Comments from the reviewers:

Reviewer #1: See attachment

Michael Dewey

Reviewer #2: The authors have performed the requested systematic review and meta-analysis, which has strengthened the main conclusions of the revised version of the manuscript. Therefore, I confirm my initial recommendation to accept it.

Reviewer #3: I reviewed manuscript 01111R2 but was not a reviewer of the initial submission of 03073.

Having examined this revised version, it is clear that the authors have addressed the main issues raised previously.

The results, which were formerly lengthy and somewhat repetitive, are now better structured, supported by summary tables and figures, and enriched by the addition of a meta-analysis.

This is the most important improvement, moving the work beyond narrative synthesis to provide quantitative pooled evidence on feasibility, diagnostic accuracy, and non-inferiority compared with conventional ultrasound.

The clinical indications are more clearly delineated, with diagnostic accuracy now stratified by anatomical structure and indication.

Risk of bias assessments, previously relegated to supplementary materials, are integrated into the main text, which strengthens the critical appraisal.

The discussion has also been broadened to include ethical, regulatory, and equity considerations, with greater attention to barriers such as infrastructure, digital literacy, and social determinants of uptake.

The roadmap for future research is more specific and operational, identifying methodological standards and clinically relevant outcomes.

Some limitations remain, especially the reliance on studies of low methodological quality and a still somewhat superficial treatment of ethical and regulatory issues, but overall the manuscript now represents a rigorous and substantially improved contribution.

It is the most comprehensive and quantitatively informed review to date on obstetric teleultrasound and is in my view suitable for publication once minor refinements are made.

Reviewer #4: The authors seek to perform a review to evaluate if teleultrasound is feasible, acceptable, diagnostically accurate and cost effective for antenatal care.

Main concern: Of the 5781 papers screened, 126 full texts were reviewed, and 61 studies were selected for inclusion. I am impressed with the number of articles that have been left out and that this process has not been performed adequately. For example, looking at Table S4, I found the work by Toscano et al "69 Testing a telediagnostic obstetric ultrasound system without a trained sonographer in a low-resource Peruvian clinic". This work was left out since it was a conference abstract. However, there is a full text article appears that can be found in:

Toscano, Marika, et al. "Testing telediagnostic obstetric ultrasound in Peru: a new horizon in expanding access to prenatal ultrasound." BMC Pregnancy and Childbirth 21.1 (2021): 328.

Furthermore, this article appears in other reviews about the same topic, including Kariman et al, a work which is cited as [3] in references of this submission. I strongly suggest the authors validate thoroughly that their inclusion/exclusion criteria have been followed correctly.

The article presents an adequate organization, presenting a table with the articles considered for the results and discussion of each section. Something that I found odd is that at the beginning of each section, the authors referenced all the considered articles. However, this list of references does not match exactly the articles presented in the table of that section. For instance, in Section 3.4.5 Three- and four-dimensional ultrasound, they mentioned they investigated 5 studies [23,39,52,60,63]. However, in Table 6, they show studies [22,23,39,52,60,63]. Same thing happens in Section 3.4.7 Low- and middle-income countries. At the beginning of the Section, they introduced studies [35,38,47,49,76,78]. However, the Table 8 shows [37,38,47,49,76,78]. Please, verify.

In addition, I believe it is also important to differentiate the different clinical applications of tele-ultrasound in antenatal care. Some of the works presented aim to give a final diagnosis, whereas others target an initial screening and adequate referral to the hospital. This will pose different requirements in equipment, connectivity, operator training, etc. Could this explain your wide range of requirements showed in equipment cost or bandwidth needed? Could this also affect the discussion of image quality?

Regarding equipment and besides the previous point, it would be important to weight the year of publication and accessibility (including cost) of the equipment in the analysis. Technology and equipment have evolved considerably from 1995 to 2024. Same consideration should be considered for bandwidth.

Regarding operator training and required skills in LMICs, besides the previous point of the target medical application, it might also be interesting to add a comment about the use of AI complementing blind sweeps for widespread use. Especially, since there is a good part of the discussion focused on this point.

---

* Please upload any figures associated with your paper as individual TIF or EPS files with 300dpi resolution at resubmission; please read our figure guidelines for more information on our requirements: http://journals.plos.org/plosmedicine/s/figures. While revising your submission, we strongly recommend that you use PLOS's NAAS tool (https://ngplosjournals.pagemajik.ai/artanalysis) to test your figure files. NAAS can convert your figure files to the TIFF file type and meet basic requirements (such as print size, resolution), or provide you with a report on issues that do not meet our requirements and that NAAS cannot fix.

After uploading your figures to PLOS's NAAS tool - https://ngplosjournals.pagemajik.ai/artanalysis, NAAS will process the files provided and display the results in the "Uploaded Files" section of the page as the processing is complete.

If the uploaded figures meet our requirements (or NAAS is able to fix the files to meet our requirements), the figure will be marked as "fixed" above. If NAAS is unable to fix the files, a red "failed" label will appear above.

When NAAS has confirmed that the figure files meet our requirements, please download the file via the download option, and include these NAAS processed figure files when submitting your revised manuscript.

FIGURES AND TABLES

SUPPLEMENTARY MATERIAL

REFERENCES

STUDY TYPE-SPECIFIC REQUESTS - SYSTEMATIC REVIEWS & META-ANALYSES

* Please report your SR/MA according to the PRISMA guidelines provided at the EQUATOR site. http://www.equator-network.org/reporting-guidelines/prisma/. Please provide the completed PRISMA checklist as Supporting Information. When completing the checklist, please use section and paragraph numbers, rather than page numbers. Please add the following statement, or similar, to the Methods: "This study is reported as per the Preferred Reporting Items for Systematic Reviews and Meta-Analyses (PRISMA) guideline (S1 Checklist)."

* Abstract: Please report your abstract according to PRISMA for abstracts (https://doi.org/10.1371/journal.pmed.1001419) following the PLOS Medicine abstract structure (Background, Methods and Findings, Conclusions). Please ensure you provide dates of search, data sources, number of studies included, types of study designs included, eligibility criteria, and synthesis/appraisal methods.

* Please note that we expect searches to be updated to within 6 months of the time of submission.

---

## [Decision Letter · Decision Letter 2]

7 Jan 2026

Dear Dr. Le Vance,

Thank you very much for re-submitting your manuscript "Teleultrasound in obstetrics: a systematic review and meta-analysis" (PMEDICINE-D-25-03073R2) for review by PLOS Medicine.

Thank you for your detailed response to the reviewers' and editors’ comments. I have discussed the paper with my colleagues, and it has also been seen again by two of the original reviewers. The changes made to the paper were satisfactory to the reviewers. As such, we intend to accept the paper for publication, pending your attention to the editors' comments below in a further revision. When submitting your revised paper, please once again include a detailed point-by-point response to the editorial comments. The remaining issues that need to be addressed are listed at the end of this email.

In revising the manuscript for further consideration here, please ensure you address the specific points made by the editors. In your rebuttal letter you should indicate your response to the editors' comments and the changes you have made in the manuscript. Please submit a clean version of the paper as the main article file. A version with changes marked must also be uploaded as a marked up manuscript file. Please also check the guidelines for revised papers at http://journals.plos.org/plosmedicine/s/revising-your-manuscript for any that apply to your paper.

We ask that you submit your revision by Feb 27 2026. However, if this deadline is not feasible, please contact me or the journal staff (plosmedicine@plos.org) by email, and we can discuss a suitable alternative.

We look forward to receiving the revised manuscript.

Sincerely,

Alexandra Tosun, PhD

Senior Editor

PLOS Medicine

plosmedicine.org

Comments from Reviewers:

Reviewer #1: The authors have addressed all my points.

Michael Dewey

Reviewer #4: Authors have addressed all my previous comments.

Requests from Editors:

As discussed via email, please update your searches to within six months of submission. We strongly encourage you to include all 2025 studies that meet your inclusion/exclusion criteria.

GENERAL

* Please confirm that your title complies with to PLOS Medicine's style. Your title must be nondeclarative and not a question. It should begin with main concept if possible. "Effect of" should be used only if causality can be inferred, i.e., for an RCT. Please place the study design ("A randomized controlled trial," "A retrospective study," "A modelling study," etc.) in the subtitle (ie, after a colon).

* Statistical reporting: Please revise throughout the manuscript, including tables and figures.

- Please report statistical information as follows to improve clarity for the reader, ""XX% (95% CI [XX,YY]; p</=)"".

- Please separate upper and lower bounds with commas instead of hyphens as the latter can be confused with reporting of negative values.

- Please repeat statistical definitions (HR, CI etc.) for each set of parentheses.

* Please ensure that all abbreviations are defined at first use throughout the text (including statistical abbreviations).

* Please ensure that tables and figures, including those in supplementary files, are appropriately referenced in the main text.

* Please review your text for claims of novelty or primacy (e.g. 'for the first time' or ‘novel’) and remove this language.

* Please confirm that any use of statistical terms (such as trend or significant) are supported by the data, and if not please remove them. The term trend should be used only when the test for trend has been conducted.

* Please define all acronyms used in each figure or table in the corresponding legend.

* Please confirm that you used patient-centered language. Please note that patient-centered language is constructed with the use of post-modified nouns putting the person first in the sentence structure.

* Please review your manuscript and edit to ensure compliance with our inclusive language requirements (https://journals.plos.org/plosmedicine/s/human-subjects-research#loc-categorization). For example, please revise language such as “rural Ethiopian women” (Table 8).

* Please remove the numbering from the headings (starting with ‘Introduction’).

ABSTRACT

* Please confirm that your abstract complies with our requirements, including providing all the information relevant to this study type https://journals.plos.org/plosmedicine/s/submission-guidelines#loc-abstract

* Please confirm that all numbers presented in the abstract are present and identical to numbers presented in the main manuscript text.

* “on a wide range of clinical outcomes” – We find this rather vague and suggest that you specify or mention a few examples, at least.

* “Identifying fetal and placental structures were frequently high using teleultrasound” – please revise for clarity.

METHODS AND RESULTS

* Thank you for providing your PRISMA checklist. Please replace the page numbers with paragraph numbers per section (e.g. "Methods, paragraph 1"), since the page numbers of the final published paper may be different from the page numbers in the current manuscript.

* Figure 2: Please note that for the last shade, the definition in the figure is ‘?10’ which we assume is supposed to be ‘<10’.

* “Four studies assessed in the first trimester, five in the second trimester, four in the third trimester, three in the first and second trimesters, 16 in the second and third trimesters, 11 in all trimesters and 20 did not disclose.” – is there a specific reason you do not cite the relevant studies here as done in the text before and after?

* Figure 7: Please note that the figure appears to be cut-off at the bottom.

* “Meta-analysis of identification rates between teleultrasound and the reference standard were available in a select number of studies for a subset of fetal and placental structures (Figure 7).” – It would be useful to mention that (only) two studies were included for meta-analysis for each subgroup.

* “Very low bandwidths (<284 Kb/s) demonstrated significantly worse image quality..” – were these statistically significant?

* “with more recent studies demonstrating a reduction in implementation cost of $10,355 - €19,356 [29,85].” – We find that presenting a range that switches between currencies is not ideal. Please revise.

DISCUSSION

* Please remove the 'conclusions' subheading from the discussion. Please also remove any other subheadings from the discussion.

General Editorial Requests

---

## [Editor Report · Decision Letter 3]

21 Jan 2026

Dear Dr Le Vance,

On behalf of my colleagues and the Academic Editor, Jennifer Elizabeth Jardine, I am pleased to inform you that we have agreed to publish your manuscript "Teleultrasound in obstetrics: a systematic review and meta-analysis" (PMEDICINE-D-25-03073R3) in PLOS Medicine.

I appreciate your thorough responses to the reviewers' and editors' comments throughout the editorial process. We look forward to publishing your manuscript, and editorially there are only two remaining points that should be addressed prior to publication. We will carefully check whether the changes have been made. If you have any questions or concerns regarding these final requests, please feel free to contact me at atosun@plos.org.

Please see below the minor points that we request you respond to:

For further transparency, we suggest the following changes to the Abstract:

*Please change to: “Overall meta-analysis demonstrated teleultrasound is non-inferior for identifying anatomical structures versus the reference standard RR 1.02 (95% CI [1.00,1.03]; n= 2 studies).”

*Please change to: “Pooled diagnostic accuracy demonstrated excellent performance, with an AUC of 0.93 (n=8 studies).”

Before your manuscript can be formally accepted you will need to complete some formatting changes, which you will receive in a follow up email (including the editorial requests above). Please be aware that it may take several days for you to receive this email; during this time no action is required by you. Once you have received these formatting requests, please note that your manuscript will not be scheduled for publication until you have made the required changes.

PRESS

Sincerely,

Alexandra Tosun, PhD

Senior Editor

PLOS Medicine